

# Detection of the freezing level with polarimetric weather radar

Daniel Sanchez-Rivas[1,*] and Miguel A Rico-Ramirez[1,*]

[1]Department of Civil Engineering, University of Bristol, Bristol, BS8 1TR, United Kingdom

**Correspondence:** Daniel Sanchez-Rivas (d.sanchezrivas@bristol.ac.uk)

**Abstract.** Accurate estimation of the Freezing Level (FL) is essential in radar rainfall estimation to mitigate the bright band enhancement, to classify hydrometeors, to correct for rain-attenuation and to calibrate radar measurements. Here we present a novel and robust FL estimation algorithm that can be applied to either Vertical Profiles (VPs) or Quasi-Vertical Profiles (QVPs) built from operational polarimetric weather radar scans. The algorithm depends only on data collected by the radar itself, and it is based on the detection of strong gradients within the profiles and relies on the combination of several polarimetric variables. The VPs and QVPs of $Z_H$ showed a good similarity in the profiles ($r \approx 0.7$) even though the QVPs are built from low-elevation angles. The algorithm is applied to one year of rainfall events and validated using measured FLs from radiosonde data. The results demonstrated that combining the profiles of $Z_H$, $\rho_{HV}$ and the gradient of the velocity $V$ showed the best FL estimation performance when using VPs, whereas combining the profiles of $Z_H$, $\rho_{HV}$ and $Z_{DR}$ showed the best FL estimation performance when using QVPs. The VP computed from the gradient of the velocity showed to be extremely valuable in the FL estimation when using VPs. The errors in the FL estimation using either VPs or QVPs are within 250 m.

## 1 Introduction

Accurate detection of the Freezing Level (FL) is important for meteorological and hydrological applications of weather radar rainfall measurements. The FL represents the height in the free atmosphere at which the measured temperature is $0°C$ and it is crucial in radar rainfall applications because it is the transition between solid and liquid precipitation which show different backscattering properties. When using weather radar data for quantitative precipitation estimation (QPE), it is necessary to apply several corrections to the reflectivity data before they can be converted into estimates of rainfall rates (Dance et al., 2019; Gourley and Hong, 2014; Mittermaier and Illingworth, 2003). For instance, corrections due to the radar Bright Band (BB) are necessary as the BB is a region of enhanced reflectivity located below the FL due to the melting of hydrometeors, that can cause an overestimation of rainfall rates (Cheng and Collier, 1993; Rico-Ramirez and Cluckie, 2007). In this case, the location of the FL is necessary to locate the BB to apply the required algorithms to mitigate the effects of this error source (Sánchez-Diezma et al., 2000; Smyth and Illingworth, 1998; Vignal et al., 1999). Above the BB, a correction for the variation of the Vertical Profile of Reflectivity (VPR) is also required, especially during stratiform precipitation, where the reflectivity of snow and ice particles decreases with height. In the UK, VPR corrections to radar data are performed by using the FL computed from a numerical weather prediction (NWP) model (Harrison et al., 2000; Mittermaier and Illingworth, 2003) assuming a 700 m BB tickness. In fact, most of the hydrometeor classification algorithms require some form of separation between liquid and solid



precipitation, hence the reliability of accurate identification of the FL (Hall et al., 2015; Kumjian, 2013a; Park et al., 2009). Even more, the attenuation of the radar signal at higher frequencies (C, X, Ka and W bands) is a significant error source for radar QPE. Attenuation correction algorithms are applied in the rain region, which again requires knowledge of the height of

the FL (Bringi et al., 1990; Islam et al., 2014; Park et al., 2005; Rico-Ramirez, 2012).

Knowledge of the FL is also useful to calibrate radar measurements. For instance, the differential reflectivity ($Z_{DR}$) is a variable that is subject to calibration errors, and the FL location is really helpful to quantify the bias of $Z_{DR}$ and mitigate errors in rain rates computed with the reflectivity ($Z_H$) and $Z_{DR}$ (Richardson et al., 2017). Depending on the radar scanning strategy, radar networks around the world have implemented operational algorithms for $Z_{DR}$ calibration. Gorgucci et al. (1999)

developed a method where vertical-looking radar observations in light rain are used to calibrate $Z_{DR}$ given that the shape of raindrops seen by the radar at $90°$ elevation is nearly circular and therefore vertical $Z_{DR}$ measurements in light rain should be around 0 dB. As vertical measurements sometimes are not available due to mechanical radar restrictions, Ryzhkov et al. (2005), Bechini et al. (2008), Gourley et al. (2009), among others, developed algorithms for $Z_{DR}$ calibration analysing the inter-dependency between $Z_{DR}$ and other polarimetric variables for several targets with a known -intrinsic value of $Z_{DR}$, e.g.

rain medium or dry snow, hence the importance of the FL estimation.

The development of polarimetric weather radar technology has allowed measuring the size and thermodynamic phase of precipitation particles. This has enabled the development of algorithms to estimate the FL directly from radar measurements based primarily on the premise that the freezing level and the melting layer (ML) are linked, as the latter describes the transition between ice/snow and water particles on the atmosphere, which show distinctive radar signatures. Using vertically pointing

radar measurements, Fabry and Zawadzki (1995) analysed the dependency of the BB on the precipitation intensity and proved the relationship between the radar BB signatures and the melting of snowflakes in stratiform precipitation. Klaassen (1988) modelled the melting layer using high-resolution Doppler radar data and found that the BB is related to the density of the ice particles for $Z_H$ and mean Doppler velocity. White et al. (2002) introduced an algorithm based on Doppler wind profiling radar scans for detecting the BB height, their results showed that the peaks of the gradients of $Z_H$, the Doppler vertical velocity

($V$) and the ML are correlated. Similarly, Baldini and Gorgucci (2006) integrated $Z_{DR}$ and the differential propagation phase ($\Phi_{DP}$) taken at vertical incidence to the analysis of the ML. They showed that the standard deviation of these measurements along with $Z_H$ and $V$ are useful for the identification of the ML using C-Band radar data.

There are different algorithms available in the literature to estimate the FL using polarimetric radar measurements. Because the BB is characterised by a region of enhanced values of $Z_H$ and linear depolarisation ratio ($LDR$) and depressions in the

correlation coefficient ($\rho_{HV}$), this has been exploited by different authors to get an estimation of the FL. Brandes and Ikeda (2004) developed an empirical procedure based primarily on idealised profiles of $Z_H$, $LDR$ and $\rho_{HV}$ that are compared with observed profiles to estimate the height of the FL; in the last step of their procedure, the height of the FL is refined using equations related to the precipitation intensity. Giangrande et al. (2008) analysed the correspondence between maxima of $Z_H$, $Z_{DR}$ and minima in $\rho_{HV}$ to estimate the boundaries of the melting layer (ML) using conventional Plan Position Indicator

(PPI) radar scans. The algorithm is tailored for scans with elevations angles between $4°$ and $10°$. Later, Boodoo et al. (2010) proposed an adaptation of this algorithm, varying the scan elevation and the range of values of $Z_H$, $Z_{DR}$ and $\rho_{HV}$ making the





algorithm more sensitive to less intense values of the BB. Matrosov et al. (2007) proposed an approach to identify rain and ML regions based only on the vertical profiles of $\rho_{HV}$ from Range Height Indicator (RHI) scans collected by an X-band radar. The method relates the depressions of the $\rho_{HV}$ profile in the ML, with the disadvantage that the absence of such depressions (e.g. warm and convective rain) hampers the application of the algorithm.

Similarly, Wolfensberger et al. (2016) designed an algorithm that combines $Z_H$ and $\rho_{HV}$ to create a new vertical profile that enables the detection of gradient variations on the vertical profiles, which are then related to the boundaries of the ML. They applied this algorithm to X-band polarimetric RHI scans, and their results showed that the algorithm is efficient to characterise the thickness of the ML and as a foundation for hydrometeor classification algorithms. Based on C-band RHI scans, Shusse et al. (2011) described the shape and variation of the melting layer on different rainfall systems and provided insights into the behaviour of $Z_{DR}$ and $\rho_{HV}$ in convective precipitation events.

Operational polarimetric weather radars are not always able to perform vertical pointing scans, or RHI scans to observe the vertical structure of precipitation. In this case, Quasi-Vertical Profiles (QVPs) can be used to monitoring the temporal evolution of precipitation and the microphysics of precipitation (Ryzhkov et al., 2016). For instance, Kumjian and Lombardo (2017) and Griffin et al. (2018) introduced new procedures for generating QVPs of the radial velocity and specific differential phase ($K_{DP}$) to explore the polarimetric signatures of microphysical processes within winter precipitation events at S-band frequencies. Similarly, Kaltenboeck and Ryzhkov (2017) analysed the evolution of the ML in freezing rain events with the QVP technique, demonstrating the ability of the QVPs to represent several microphysical precipitation features as the dendritic growth layer and the riming region, both of them related to the ML and FL. Despite the enormous benefits that QVPs bring in terms of improving our understanding of the microphysics of precipitation, there is very little research on the use of QVP-based algorithms to estimate the FL.

Most of the algorithms mentioned above require measurements of the vertical structure of precipitation (e.g. RHI scans) to estimate the FL, which might not be available in operational weather radar networks. Operational weather radar networks produce volume scans at different elevation angles that allow an insightful analysis of the characteristics of the FL. Therefore, this paper presents a novel, automated, operational and robust algorithm that can accurately detect the FL based on QVPs or Vertical profiles (VPs) of polarimetric measurements collected from an operational C-band dual-polarisation weather radar and demonstrate its performance for different types of events. The results are compared with FL heights from the UKMO high-resolution radiosonde data. The latter enables the analysis of the performance of QVPs and VPs on the FL estimation. Note that the proposed algorithm is not intended to replace NWP-based FL estimation methods, but it is an alternative way to estimate the FL when only polarimetric weather radar measurements are available.

## 2   Datasets and Methods

The Chenies C-band operational weather radar located in South East England was selected for this work as it was one of the first UKMO radars upgraded with polarimetric capabilities (Norman et al., 2014) (see Table 1 and Figure 1). The radar transmits both horizontal and vertical pulses simultaneously and receives co-polar signals at the same time, generating polarimetric



measurements of $Z_H$, $Z_{DR}$, $\rho_{HV}$ and $\Phi_{DP}$. Doppler velocity $(V)$ information on the observed droplets is also available (Met Office, 2013). LDR measurements are also produced using a special separated pulse. The scanning strategy implemented on this radar generates different products:

- 5 PPI scans sampled on Long Pulse (LP) mode (pulse length= $2,000\mu s$; range covered=250 km) at 0.5°, 1°, 2°, 3° and 4° elevation angles with a 600 m gate resolution every 5 minutes.

- 5 PPI scans sampled on Short Pulse (SP) mode (pulse length= $500\mu s$; range covered=115 km) at 1°, 2°, 4° 6° and 9° elevation angles, every 10 minutes and same gate resolution as above.

- One SP PPI scan at vertical incidence (range covered=12 km) every 10 minutes with 75 m gate resolution.

- One PPI scan with LDR measurements every 5 minutes at the lowest elevation (0.5°).

Radiosonde data were used to validate the FL estimated from radar observations. The radiosonde is an instrument that is
released into the atmosphere to measure several atmospheric parameters. The UKMO uses the Vaisala RS80 radiosonde model to collect upper-air observations twice daily at different locations across the UK. The ascent of the radiosonde extends to heights of approximately 10-30 km and take measurements at 2-second intervals (Met Office, 2007). The closest station to the Chenies radar (see description below) is the Herstmonceux station (see location in Figure 1), which provides high-resolution radiosonde information of vertical profiles of pressure, temperature, relative humidity, humidity mixing ratio, sonde position,
wind speed and wind direction. As temperature measurements provide insights for the FL location and features, this data was processed in a height-versus-temperature and time-versus-temperature format to enable the location of the $0°C$ isotherm and evaluate the algorithm performance.

**Table 1.** Chenies radar characteristics.

| Chenies radar | |
|---|---|
| Location | 51.6892, -0.5297 |
| Wavelength | $\lambda = 5.3\ cm$ |
| Multiple elevation scans | 0.5° to 90° |
| Beam-width | 1.0° |
| PRF | 900 $Hz$ (SP) - 300 $Hz$ (LP) |
| RPM | 3.6 (SP) - 1.4 (LP) |

Polarimetric scans related to precipitation events throughout 2018 were analysed for the design and evaluation of the algorithm. To reduce the probability of ground clutter contamination and beam spreading effects, only SP scans from the 4°,
6°, 9° and 90° elevations angles were retained for further processing. A total of 94 rainfall events with visible signatures of enhancement (related to the FL) on the polarimetric variables were selected, of which 25 had a suitable temporal matching between the precipitation events recorded by the radar and the data collected by the radiosondes. Then, a pre-processing of the raw-radar-data is carried out to discard non-meteorological echoes and construct the profiles of polarimetric variables:



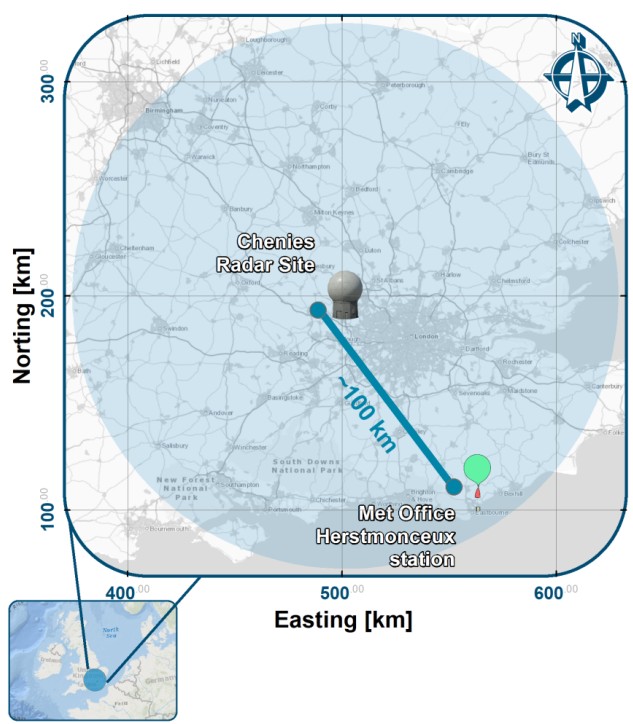

**Figure 1.** Location and coverage (on SP mode) of the Chenies radar and location of the Herstmonceux station.

– For the $4°$, $6°$ and $9°$ elevations scans, remnant clutter and anomalous propagation echoes were removed using the
algorithm proposed by Rico-Ramirez and Cluckie (2008), specifically calibrated with this radar data. Then, following
the procedure suggested by Ryzhkov et al. (2016), we generated QVPs of $Z_H$, $Z_{DR}$, $\rho_{HV}$ and $\Phi_{DP}$ measurements.
The procedure suggest the azimuthal averaging of the polarimetric measurements at high elevation scans ($10°$-$30°$),
and results in height-versus-time representations of the polarimetric variables that enable the observation of the vertical
structure of precipitation and the identification of distinctive ML signatures.

– For the vertical scans, the first kilometre of data is not usable due to some inherent radar limitations, e.g. the de-ionisation
time of the Transmit-Receive (TR) cell (Darlington, 2019, personal communication). After discarding the data below
this height, an azimuthal averaging of the polarimetric and Doppler velocity data collected at vertical incidence was
performed, generating the VPs of $Z_H$, $Z_{DR}$, $\rho_{HV}$, $\Phi_{DP}$ and $V$. Based on the profiles of vertical velocity $[V]$, we propose
a new variable: $[gradV]$, that calculates the gradient of the $90°$ velocity profile. This new variable accentuates the profile
extremes that are related to the change in the hydrometeor fall velocities from ice/snow to rain. The gradient of $V$ is
computed using first-order central differences in the interior points and first-order forward or backwards differences at
the boundaries; for an in-depth description of numerical differentiation and finite-differences methods see Moin (2010).

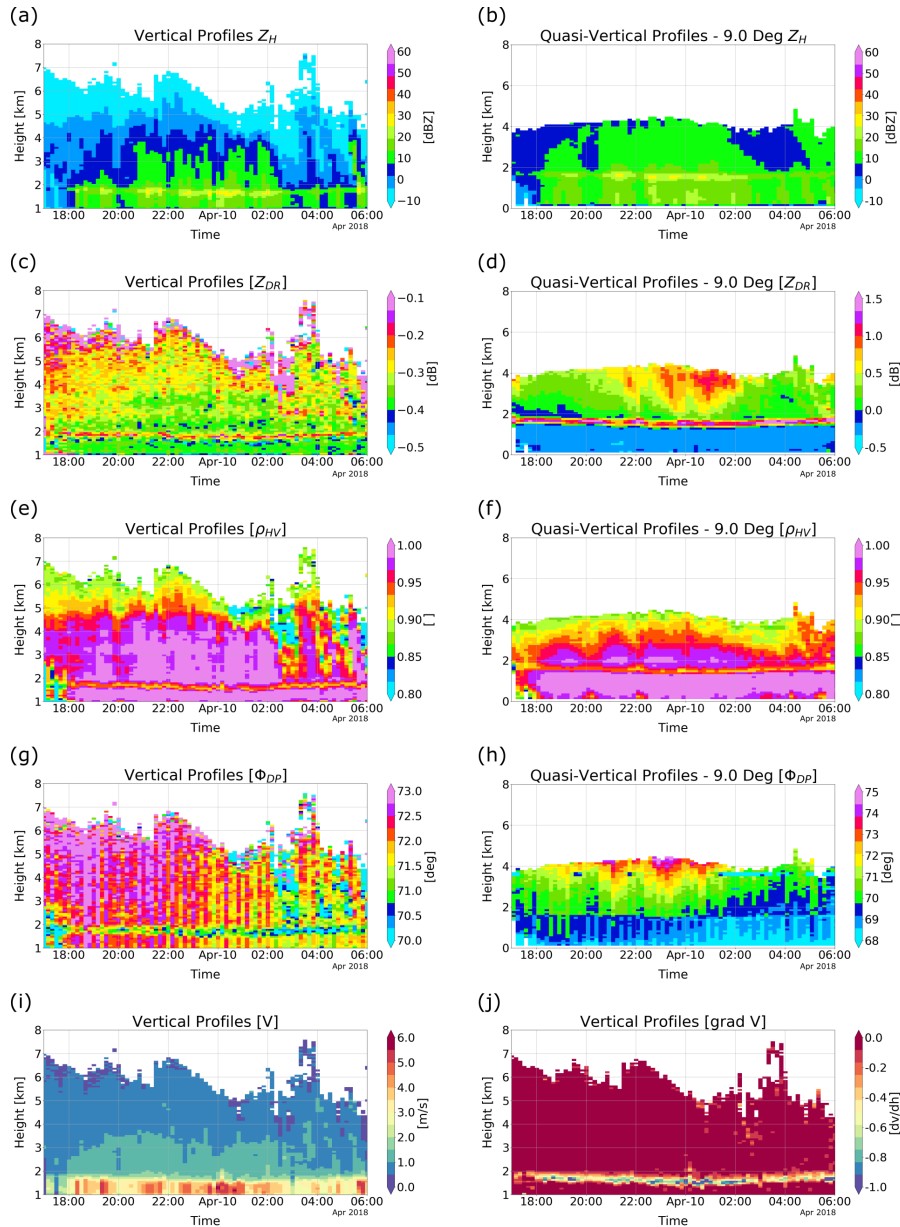

**Figure 2.** HTI plots of $Z_H$ (a-b), $Z_{DR}$ (c-d), $\rho_{HV}$ (e-f) and $\Phi_{DP}$ (g-h) generated from VPs (left) and QVPs (right) for a precipitation event recorded by a weather radar located at Chenies, UK. Also, (i) portrays the Doppler vertical velocity of hydrometeors, whilst (j) shows a plot of the profiles based on the gradient of V measurements $[dV/dh]$.



## 3 Polarimetric signatures of the melting layer

The raw-polarimetric radar data were pre-processed as described in the previous section in order to generate height-versus-time

plots of VPs and QVPs of all polarimetric measurements available to visualise the ML signatures of different precipitation events. Figure 2 displays an example for a stratiform rainfall event recorded between 9-10 April 2018 using VPs and QVPs at $9°$ elevation angle. It can be seen that every radar variable exhibits distinctive features that provide unique information for the identification of the melting layer and the FL on both, VPs and QVPs, e.g. Figures 2a-2b, and 2c-2d exhibit regions of enhanced values of $Z_H$ (BB) and $Z_{DR}$, respectively, that are visible just below 2 km in height. Concurrently, Figures 2e-2f

and 2g-2h show that $\rho_{HV}$ and $\Phi_{DP}$, are sensitive to the phase and shape of hydrometeors, while Figure 2i shows that the fall velocities of snow particles are lower compared to rain particles, which is an important feature that can be used to detect the FL. Figure 2j shows the enhanced region where the transition between the fall velocities of snow and rain particles occurs in the proposed new variable $[gradV]$. The different signatures expected in the ML on the QVPs and VPs are explained next.

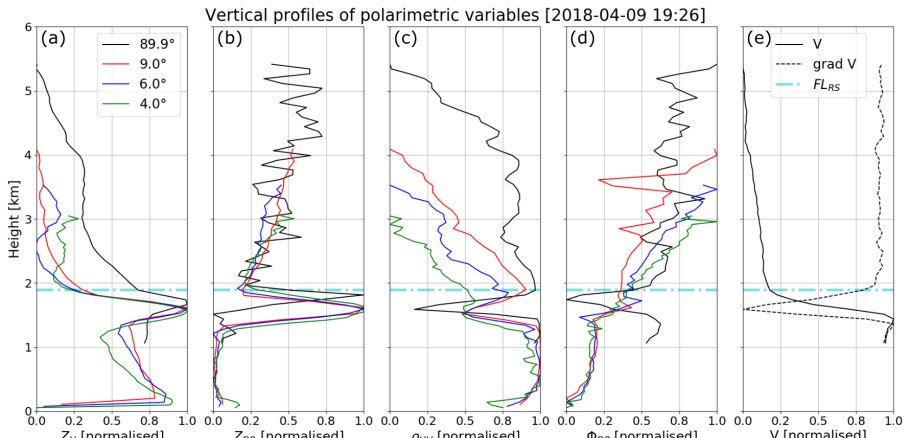

**Figure 3.** Normalised version of VPs and QVPs generated from polarimetric scans recorded at different elevation angles, related to a stratiform-type rain event.

The VPs and QVPs are normalised to enhance the features observed in the BB and at the FL. An example of normalised

polarimetric profiles related to a stratiform event, corresponding to different elevation angles and with the extents set for comparison purposes only, is shown in Figure 3 along with the closest-in-time radiosonde data where the temperature reaches $0°C$ $(FL_{RS})$. Given that the main objective of this work is to estimate the FL based on the geometric features of the polarimetric profiles, herein, we will try to explain the enhancements that the ML bring-up into the variables. Figure 3 shows that the enhancement on the polarimetric variables is related to the variation of the phase and concentration of the hydrometeors, i.e.

the ML, and the upper boundary of these enhancements (i.e. the BB top) is related to the FL, just below 2 km in altitude. This enhancement is not necessarily at the same height in all polarimetric variables, but this has to do with the backscattering properties of the particles and their relationship with the measured variable, as well as the methods used in the construction of





the profiles, as demonstrated by Brandes and Ikeda (2004), Fabry and Zawadzki (1995) and Ryzhkov et al. (2016). Moreover, the QVPs provide the information below 1 km in elevation; this is important for the analysis of showers or events with freezing

levels at relatively low altitude.

The reflectivity ($Z_H$) represents the power backscattered by precipitation particles, thus providing a general idea of the concentration, size, phase and water content of the hydrometeors (Gourley and Hong, 2014). Figures 2a and 2b demonstrates that, to a certain extent, the values of $Z_H$ represent similar backscattered energy taken either at vertical or horizontal elevations, this will be subject of analysis later on. It also can be seen on both, QVPs and VPs, the well-known BB effect on $Z_H$, that

yields a peak related to enhanced values of reflectivity due to the difference in the dielectric factor for ice and water (Islam and Rico-Ramirez, 2014). As shown in Figure 3a, the BB is easily observed in stratiform events (close to 1.7 km); however, its boundaries do not always match the extents of the ML. Whilst for convective events, the profiles of $Z_H$ do not show a sharp BB feature, and therefore the boundaries of the ML for this type of events are more challenging to identify.

The differential reflectivity ($Z_{DR}$) represents the difference between the horizontal and vertical reflectivity values $[Z_{(h,v)}]$,

depending on the orientation, shape and density of the hydrometeors (Islam and Rico-Ramirez, 2014); therefore, ($Z_{DR}$) measurements for the QVPs and VPs may represent different features in this variable. From Figure 2c we can observe that $Z_{DR}$ is not calibrated, as we expect near-to-zero values of this variable on the rain region for vertically pointing measurements as raindrops are symmetrical on average when observed from underneath. Hence, $Z_{DR}$ measurements used in this work present a bias that requires calibration if intended for use in radar QPE, but we did not attempt to calibrate $Z_{DR}$ scans because this

requires the knowledge of the freezing level, which is something we want to estimate first. For both QVPs and VPs, $Z_{DR}$ profiles show similar behaviour for stratiform events: Figure 3b shows that $Z_{DR}$ exhibit mean small o null slope changes on the rain medium (below 1.2 km), but there is a noticeable peak associated with the ML on both VPs and QVPs (at a similar height of the BB), and the top and bottom boundaries are easy to discern. Brandes and Ikeda (2004) and Ryzhkov et al. (2016) showed that the presence of different phase scatters, or melting, randomly oriented ice particles within the melting layer and

the mixing of hydrometeors produce the peaks in $Z_{DR}$ often present in stratiform events. However, for profiles related to convective events (not shown), the VPs sometimes exhibit an inverse peak exactly above the rain medium and then generating a noisy, random pattern on the melting layer, that makes the estimation of the FL more difficult using VPs of $Z_{DR}$. Finally, the most significant difference for this variable can be seen in Figures 2c and 2d where the values of $Z_{DR}$ for VPs and QVPs are not the same, regardless of the hydrometeor phase.

The correlation coefficient ($\rho_{HV}$) measures the correlation between $Z_H$ and $Z_V$ measurements, and it is sensitive to the distribution of particle sizes and shapes, hence being less sensitive to the type of precipitation rather than to the phase of the hydrometeors (Gourley and Hong, 2014). Figure 3c shows that the melting layer causes a similar response on $\rho_{HV}$ as in $Z_H$ and $Z_{DR}$, but in the opposite direction, resulting in a depression on the profiles at 1.5-1.6 km in height for QVPs and VPs, respectively. This depression is the result of the shift between high values of $\rho_{HV}$, related to raindrops and ice crystals and

lower values triggered by the variety of shapes and axis ratios related to large, melting hydrometeors (Kumjian, 2013b). The behaviour of $\rho_{HV}$ is similar on both, VPs and QVPs from $9°$ elevation, for stratiform or convective events, where the major difference lies in the depth of the depressions. However, again, this may be caused by the resolution and elevation angle of





the original scans (Ryzhkov et al., 2016). On the other hand, the QVPs constructed from lower elevation angles, i.e. $4°$ and $6°$ exhibit less pronounced peaks related to the ML and a pronounced decrease rate above 1.8 km that difficult the location of

the FL. Additionally, $\rho_{HV}$ is a reliable indicator of the quality of the radar data as $Z_H$, and $Z_V$ are not equal, especially when measuring snow/ice or melting ice particles, however, in the rain medium their correlation is close to 1, becoming an indicator of the quality of the polarimetric radar measurements (Kumjian, 2013a) and then tends to decrease slightly for large oblate raindrops. Figures 2e and 2f show that $\rho_{HV}$ is close to 1 within the rain region, this is a good indicator of the quality of the datasets.

As can be seen in Figures 2g, 2h, and 3d, the signatures of the ML on the differential propagation phase $(\Phi_{DP})$ are, to a certain degree, ambiguous, especially on the QVPs. $\Phi_{DP}$ is a cumulative variable that represents the difference in phase between horizontal and vertical electromagnetic waves, providing valuable information about the shape and concentration of the particles (Islam and Rico-Ramirez, 2014). Hence, the peaks on this type of profiles are related to a greater concentration of particles, e.g. accumulation of melting or snowflake particles that may be related to the ML or the dendritic growth layer (DGL) as

demonstrated by Griffin et al. (2018) and Ryzhkov et al. (2016). Figure 3d shows that the QVPs of $\Phi_{DP}$ from $9°$ elevation, exhibit a small peak at 1.7 km in height related to the backscatter differential phase, but is not as pronounced as with the other polarimetric variables, although there are significant peaks aloft (between 2.8 and 3.8 km), that may represent particle (ice or snowflakes) alignment on the DGL, as suggested by Kaltenboeck and Ryzhkov (2017), while lower elevation angles do not show strong signatures on the ML nor the DGL. In contrast, for $90°$ elevation scans, there is a well-defined depression in $\Phi_{DP}$

related to melting and particle growth (Brandes and Ikeda, 2004) at 1.8 km in height that closely matches the height of the BB; regarding the signatures of the DGL on the VPs, due to the noisiness of the profile above the ML it is difficult to determine if these peaks are related to the DGL.

Last but not least, Figures 2i and 3e shows the profiles related to the Doppler vertical velocities $(V)$ and the signatures of the ML on this variable. It can be seen that the fall velocity $[\mathrm{ms}^{-1}]$ of the hydrometeors is relatively constant and close to zero

above the ML, which is related to the fall velocity of ice and snow particles, then there is a sharp increase in the fall velocity of the precipitation particles in the melting layer, and then becomes constant again in the rain region, having said that, it is difficult to incorporate this variable into an algorithm because their features are not easy to identify by using an automated procedure. Conversely, the VP of the gradient of V $(gradV)$, shown as the dotted line in Figure 3e, exhibits a shape and peak similar to the rest of polarimetric variables, where the upper and lower curvatures of the peak match closely the top and bottom extents

of the ML.

## 4   Algorithm to identify the freezing level

The freezing level algorithm (FLA) has been designed for the automatic detection of the FL using either QVPs or VPs, under the premise that the peaks on each polarimetric profile and their curvatures are related to the ML, and the upper boundary of the ML is related to the height of the FL. The FLA exploits the procedure proposed by Wolfensberger et al. (2016) that combines

$Z_H$ and $\rho_{HV}$ to create a new profile to enhance the ML features. However, it is clear from Figure 3 that there are additional





variables, such as $[gradV]$, that may improve the identification of the FL. Therefore, the proposed algorithm is designed to include all the radar variables available and analyse their relevance on the estimation of the FL. Some considerations are made for its design, e.g. to minimise the effect of beam broadening the analysis is constrained to a height of 5 kilometres. Also, as shown in Figure 3, some profiles get noisy above the FL or contain spurious echoes aloft, making necessary to set an initial

upper extent for the algorithm to work. The FLA is divided into two parts. The first part determines if the profile contains elements to detect the FL based on the combination of two profiles and setting an upper limit for its implementation. The second part estimates the FL based on a combination of the polarimetric profiles and their features. The algorithm uses either QVPs or VPs, but we avoid combining both profiles as VPs might not be available in other weather radar networks. A flowchart that illustrates the steps of the FLA is shown in Figure 5 and described below.

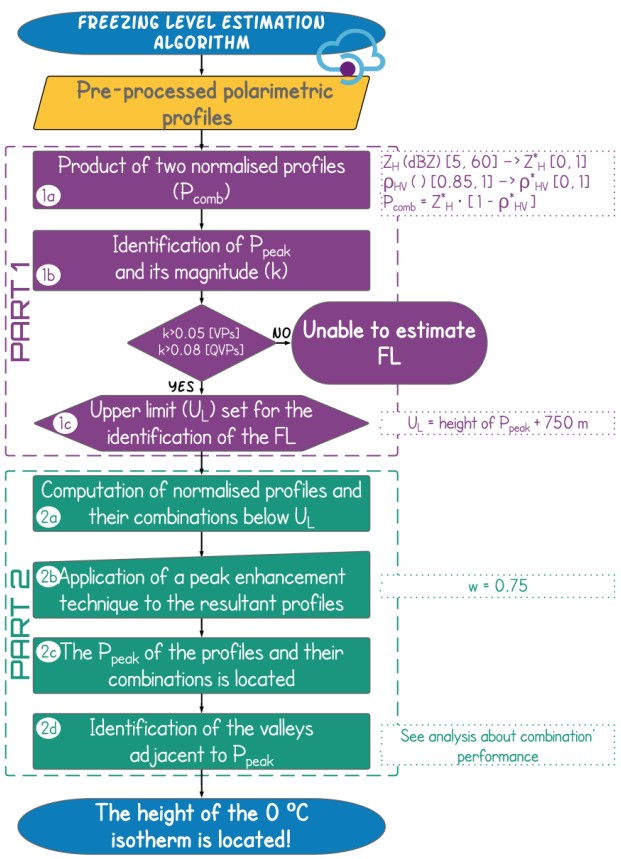

**Figure 4.** Flowchart of the proposed FLA.

1. **Part 1**

   The first part of the FLA identifies profiles that are likely to contain signatures related to the ML and set an upper limit in the profiles to make use of all the available variables.



1.a The algorithm takes advantage of the distinctive signatures on the profiles of $Z_H$ and $\rho_{HV}$ on both, VPs and QVPs, to perform an initial identification of rain echoes. These two profiles are normalised and combined into a single profile $(P_{comb})$ as suggested by Wolfensberger et al. (2016), but using different thresholds that are related to drizzle, heavy rain, snow and ice (Kumjian, 2013a; Fabry, 2015). The values of $Z_H$ between 5 and 60 dBZ are normalised between 0 and 1 $[Z_H(dBZ)[5,60] \rightarrow Z_H^*[0,1]]$, whereas the values of $\rho_{HV}$ between 0.85 and 1 are normalised between 0 and 1 $[\rho_{HV}(\,)[0.85,1] \rightarrow \rho_{HV}^*[0,1]]$. Note that (*) in the polarimetric variable indicates a normalised variable. Values outside these intervals are fixed to 0 and 1, correspondingly. The combination of these normalised profiles can be expressed as:

$$P_{comb} = Z_H^* \cdot (1 - \rho_{HV}^*) \tag{1}$$

1.b The FLA locates the peaks on the profile $P_{comb}$ comparing neighbouring values: a peak is a sample whose direct neighbours have smaller magnitudes. Using this procedure, the inverse peaks within the profile can be defined as the valleys of $P_{comb}$; this enables the computation of the magnitude $(k)$ of the peaks i.e. the horizontal distance between the peak and its lowest valley. The peak with the higher magnitude is set as $P_{peak}$. If the magnitude $(k)$ of $P_{peak}$ is less than 0.05 for VPs or 0.08 for QVPs, the FLA determines that the gradients are not strong enough to correspond to ML signatures and therefore the profile does not contain elements to detect the FL. Further discussion on the values of these parameters is provided in the next sections.

1.c Otherwise, an upper limit $(U_L)$ is set taking the height of $P_{peak}$ and adding 750 meters above. This value is selected to locate the top of the ML from the ML peak. Usually the ML thickness (i.e. between BB bottom and BB top) can reach values less than about 800 m (Fabry and Zawadzki, 1995), hence 750 m is sufficient to locate the top of the ML. Figure 5a illustrates the first step of the algorithm.

2. **Part 2**

Once $U_L$ is defined, the algorithm identifies the $0°C$ height based on a profile that is the result of the combination of several polarimetric profiles as follows.

2.a The profiles of the available polarimetric variables have an upper limit set by $U_L$ to search for the FL. Every individual profile is normalised. Note that $Z_H^*$ and $\rho_{HV}^*$ were already normalised in step 1.a, whereas the rest of the variables are normalised using the minimum and maximum values of each variable. A new profile is computed following Equation 2 for VPs or Equation 3 for QVPs:

$$P_i^* = (1 - gradV^*) \cdot (Z_H^*) \cdot (Z_{DR}^*) \cdot (1 - \rho_{HV}^*) \cdot (1 - \Phi_{DP}^*) \tag{2}$$

$$P_i^* = (Z_H^*) \cdot (Z_{DR}^*) \cdot (1 - \rho_{HV}^*) \cdot (\Phi_{DP}^*) \tag{3}$$

where $i$ depends on the combination of the variables used according to Table 2.





2.b $P_i^*$ will very likely show a peak within the ML. To refine the boundaries of this peak i.e. the valleys adjacent to the peak, the FLA applies a peak-enhancement technique expressed by:

$$P_i = P_i^* - (w \cdot P_i^{*''}) \tag{4}$$

Where $P_i$ is the enhanced profile, $P_i^*$ is the profile given by Equations 2 or 3, $w$ is a weighting factor and $P_i^{*''}$ is the second derivative of $P_i^*$. The optimum choice of the parameter $w$ depends upon the signal-to-noise ratio and the desirable sharpening extent. Table 2 lists the enhanced profiles produced by combining different polarimetric profiles, and Figure 5b shows the enhancement of the peak and valleys. Details on the choice of the parameter $w$ are presented in the following sections.

2.c For each profile $P_i$, the magnitude $(k)$ of $P_{peak}$ is computed as in step 1.b. The same threshold [$k < 0.05$ for VPs; $k < 0.08$ for QVPs] is applied in this step.

2.d Figure 5b shows that the valleys of the profile $P_i$ are related to the boundaries of the ML. These valleys can be placed by searching the inverse peaks directly above and below $P_{peak}$. Finally, the algorithm allocates the top valley of $P_i$ as the estimated height of the FL $(FL_e)$.

**Table 2.** Possible combinations of polarimetric variables for VPs and QVPs used on the FL estimation.

| [VPs] | [QVPs] | $P_1$ | $P_2$ | $P_3$ | $P_4$ | $P_5$ | $P_6$ | $P_7$ | $P_8$ | $P_9$ | $P_{10}$ | $P_{11}$ | $P_{12}$ | $P_{13}$ | $P_{14}$ | $P_{15}$ |
|---|---|---|---|---|---|---|---|---|---|---|---|---|---|---|---|---|
| $1-gradV^*$ | - | ○ | ○ | ○ | ○ | ○ | ○ | ○ | ○ | ○ | ○ | ○ | ○ | ○ | ○ | ○ |
| $Z_H^*$ | $Z_H^*$ | ○ | ○ | ○ | ○ | ○ | ○ | ○ | ● | ● | ● | ● | ● | ● | ● | ● |
| $Z_{DR}^*$ | $Z_{DR}^*$ | ○ | ○ | ○ | ● | ● | ● | ● | ○ | ○ | ○ | ○ | ● | ● | ● | ● |
| $1-\rho_{HV}^*$ | $1-\rho_{HV}^*$ | ○ | ● | ● | ○ | ○ | ● | ● | ○ | ○ | ● | ● | ○ | ○ | ● | ● |
| $1-\Phi_{DP}^*$ | $\Phi_{DP}^*$ | ● | ○ | ● | ○ | ● | ○ | ● | ○ | ● | ○ | ● | ○ | ● | ○ | ● |

| [VPs] | [QVPs] | $P_{16}$ | $P_{17}$ | $P_{18}$ | $P_{19}$ | $P_{20}$ | $P_{21}$ | $P_{22}$ | $P_{23}$ | $P_{24}$ | $P_{25}$ | $P_{26}$ | $P_{27}$ | $P_{28}$ | $P_{29}$ | $P_{30}$ | $P_{31}$ |
|---|---|---|---|---|---|---|---|---|---|---|---|---|---|---|---|---|---|
| $1-gradV^*$ | - | ● | ● | ● | ● | ● | ● | ● | ● | ● | ● | ● | ● | ● | ● | ● | ● |
| $Z_H^*$ | - | ○ | ○ | ○ | ○ | ○ | ○ | ○ | ○ | ● | ● | ● | ● | ● | ● | ● | ● |
| $Z_{DR}^*$ | - | ○ | ○ | ○ | ○ | ● | ● | ● | ● | ○ | ○ | ○ | ○ | ● | ● | ● | ● |
| $1-\rho_{HV}^*$ | - | ○ | ○ | ● | ● | ○ | ○ | ● | ● | ○ | ○ | ● | ● | ○ | ○ | ● | ● |
| $1-\Phi_{DP}^*$ | - | ○ | ● | ○ | ● | ○ | ● | ○ | ● | ○ | ● | ○ | ● | ○ | ● | ○ | ● |

*Note:* * refers to the normalised version of the variables.

As can be seen in Figure 5a, the combination of $Z_H^*$ and $1-\rho_{HV}^*$ produces a profile with a peak $(P_{peak})$ related to the ML that is useful to estimate the FL. The magnitude $(k)$ of $P_{peak}$ is important to discard profiles that do not have a strong enhancement related to the ML. However, additional variables can be used (see Equations 2 and 3) to refine the estimation of the FL . The upper limit $(U_L)$ allows the use of other variables that otherwise could not be part of the algorithm due to noisiness





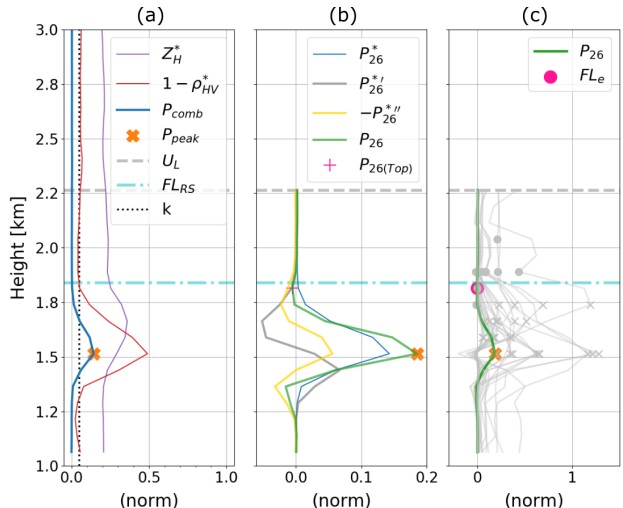

**Figure 5.** Depiction of the implementation of the algorithm for the FL estimation.

or spurious echoes present at the top of the profiles. Figure 5b shows the importance of the refinement of the profile, e.g. the profile combination $P_{26}^* = (1 - gradV^*) \cdot (Z_H^*) \cdot (1 - \rho_{HV}^*)$ has a peak related to the ML and the valley located at the top of this peak is close to the FL, but it is difficult for a peak-detection algorithm to detect its height as it is not as pronounced as required. The use of the first derivative of the profile, i.e. $P_{26}'$, is not helpful as the peaks are not close to the FL. The profile

$P_{26}$ results from the implementation of Equation 4, where a value of $w = 0.75$ enhance the peak and its valleys enough for the algorithm to detect their location. A proper choice of the parameter $w$ depends on the desired weight to the original profile rather than its second derivative. The impact of the parameters $k$ and $w$ into the algorithm will be discussed in the following sections.

## 5   Results

### 5.1   VP and QVP comparison

To support the performance and outputs of the algorithm is necessary to assess the consistency between the available radar data. Given that the polarimetric variables portray the echoes depending on the elevation angle where they were taken, it is only possible to compare QVPs and VPs of $Z_H$, as these variables measure similar properties of the raindrops taken either at low-elevation angles ($4°$, $6°$ and $9°$) or 90-deg elevations respectively. From the total of 94 rainfall events, we carried out a

manual classification of rain events according to the findings of Fabry and Zawadzki (1995) and Rico-Ramirez et al. (2007). A total of 68 events were classified as stratiform. This category includes low-level rain and rain with BB as they showed the well-known enhancement of reflectivity observed within the ML or look-alike drizzle events below the $0°C$. On the other hand, 26 events recorded mainly during the summer met the characteristics of showers, i.e. indistinguishable signatures of the ML





in the $Z_H$ profiles, in which higher values of reflectivity are present; the latter is the type of precipitation that is less common
in the UK (Collier, 2003). The comparison between VPs and QVPs takes into account the time stamp and spatial resolution of
the profiles. The Pearson correlation coefficient ($r$) is computed to analyse the consistency between the VPs and QVPs. The
results for stratiform and convective events are shown in Figures 6 and 7, respectively.

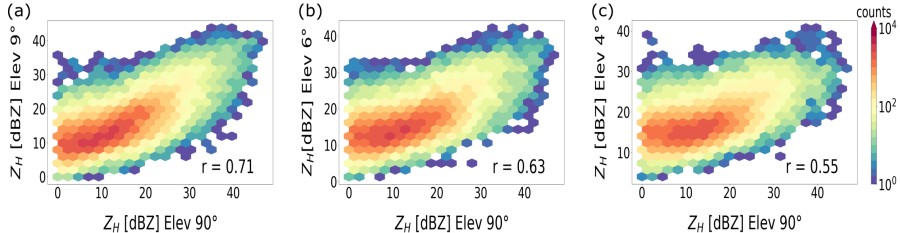

**Figure 6.** Comparison of VPs and QVPs of $Z_H$ generated at three elevation angles for a collection of stratiform events. Counts indicate the
number of points in the hexagon

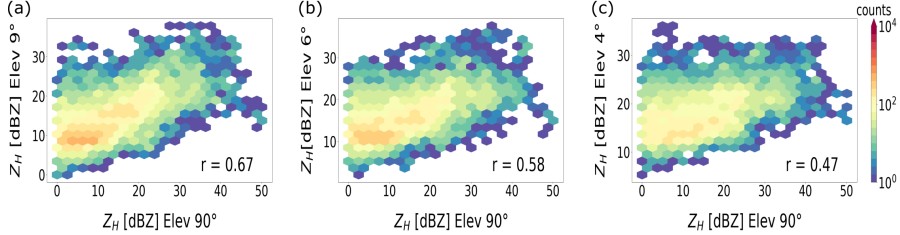

**Figure 7.** As in Fig. 6, but for a collection of convective events. Counts indicate the number of points in the hexagon

Figure 6a shows that the agreement between VPs and QVPs (constructed from $9°$ scans) resides on relative low values of
reflectivity, that are most probably related to light and moderate rain rates; this is expected for stratiform-type events. Figures
6b and 6c show that the agreement diminishes when decreasing the elevation angle, whilst higher values of $Z_H$ do not always
find their pairs as the elevation decrease. This could be explained by the averaging process carried out in the construction
process of the profiles, as the radar resolution volume increases with distance. On the other hand, Figure 7 shows a more
scattered distribution of $Z_H$ for shower-type events. Higher values of $Z_H$, related to moderate to heavy rain-rates are present.
Again, the correlation decreases for lower elevation angles, and it can be seen that there are mismatches for cells with higher
values of reflectivity.

It is worth mentioning that a similar analysis was carried out using other polarimetric variables, but as expected, the results
were not consistent as only $Z_H$ describe similar properties of the precipitation measurements taken either at horizontal or
vertical elevation angles.



## 5.2 Implementation of the FL algorithm

As described in section 4, the FLA performs a pre-classification of profiles that are likely to contain signatures related to the ML. Some tests were carried out by replacing $Z_H^*$ with other variables (e.g. $Z_{DR}$ or $1 - gradV^*$) to see if there is any improvement in the pre-classification. From Figure 3b it is clear that the QVPs of $Z_{DR}$ exhibit a pronounced peak related to the ML even for low elevation angles, but unfortunately $Z_{DR}$ is not calibrated and the thresholds for normalising this variable may vary depending on the elevation angle. On the other hand, replacing $Z_H^*$ with the profile $1 - gradV^*$ for the VPs could

improve the pre-classification, but this may restrict the implementation of the algorithm, i.e. it would be only applicable if vertical velocity profiles are available. Although we observed some improvements using these variables in the first part of the FLA, especially for convective events, we wanted to keep this part as simple and robust as possible to avoid a complex algorithm. Hence we used the combination of $Z_H^*$ and $\rho_{HV}$ for part 1 of the algorithm as initially proposed by Wolfensberger et al. (2016).

Also, as shown in Figures 4, 5a, 5b and Equation 4, the algorithm relies on the parameters $k$ and $w$. These parameters can be adjusted according to the radar datasets, e.g. the parameter $k$ can be affected by the quality of $\rho_{HV}$: in our datasets and after the removal of non-meteorological echoes, $\rho_{HV}$ exhibits values close to 0.85 in the ML, on both QVPs and VPs, but this may vary depending on the type of radar, scanning strategy and quality of the datasets. We set $k = 0.05$ for VPs and $k = 0.08$ for QVPs empirically, and these values allow the algorithm to discard enhancements in the profile not strong enough

to be related to the ML. On the other hand, Equation 4 is applied to the profiles to enhance the peaks (and valleys) within the profile, thus refining the detection of the FL. This equation combines the original profile with its second derivative, that can be weighted with the parameter $w$. As shown in Figure 5b, the second derivative of the profile exhibits deeper peaks, but its top boundary is still far from the measured FL. After several trials, we set $w = 0.75$ as this value proves to enhance the peaks of the original profile without compromising the match of the top boundary and hence, improving the estimation of the FL. Again,

this parameter can be adjusted depending on the radar datasets, e.g. profiles that exhibit smoother peaks due to the nature of its construction process and the resolution of the original scans, or profiles with vertical resolution too coarse can be adjusted with the parameter $w$ for a better performance of the algorithm.

## 5.3 FL estimation from VPs

The performance of the FL estimation from VPs is compared against the FL measured by radiosonde data over one year of

rainfall events. Since soundings are released twice daily, the radiosonde data is extended at several time-steps to create short time-windows and enable a comprehensive comparison with the radar data. Performance metrics [Pearson correlation coefficient ($r$), Mean Absolute Error ($MAE$), Root Mean Square Error ($RMSE$)] between the measured ($FL_{RS}$) and estimated ($FL_e$) FL heights are computed. Figure 8 shows the results for a 60-min window, i.e. the measured ($FL_{RS}$) is set as constant for 30-min before and after the time stamp of the radiosonde.

Figure 8 shows the influence of all polarimetric variables on the estimation of the FL. In Figure 8a, the variable *n profiles* is an indicator of the additional false-positive cases that can be detected by each combination, this takes place due to the presence





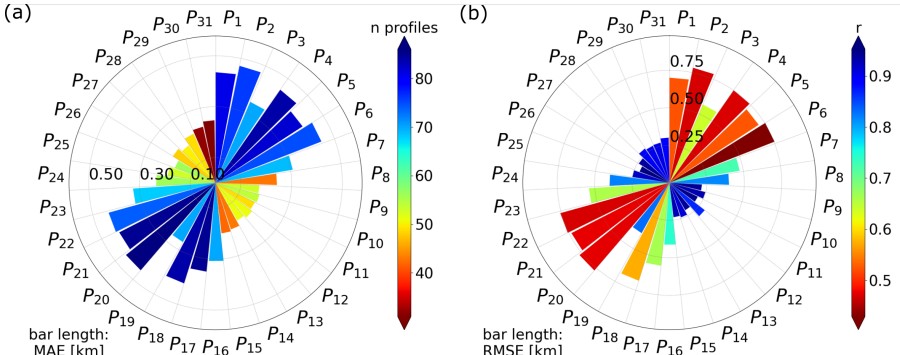

**Figure 8.** Errors in the FL estimation for VP using a ±30 minute window. In (a) the bar length represents the MAE (in km) for every polarimetric combination and colour represents the number of vertical profiles analysed for every polarimetric combination; in (b) the bar length represents the RMSE (in km) and colour represents r.

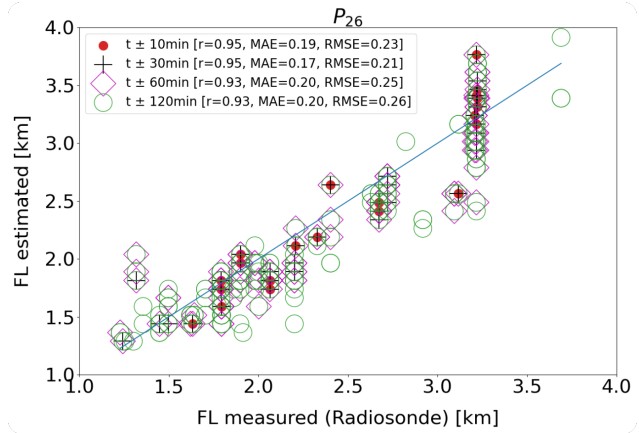

**Figure 9.** Heights of the $0°C$ isotherm measured by radiosonde versus FL estimated by the algorithm using the combination $P_{26}$ for several time windows. The 1:1 line is shown in blue. MAE and RMSE in km.

of peaks not related to the ML but strong enough to surpass by the first part of the algorithm, e.g. profiles that contain only one variable or combinations that engage $\Phi_{DP}$ into the analysis, where strong peaks are found but not associated with the ML. Overall, Figure 8 shows that the combinations that include $Z_H^*$, $[1-\rho_{HV}^*]$ or $[1-gradV^*]$ improve the accuracy of the FLA

e.g. $P_9$, $P_{11}$ or $P_{26}$, as the correlation and the errors are relatively low for these combinations. After a visual assessment of the performance of each profile and in combination with the statistics computed above, we determine that the profile combination $P_{26} = [Z_H^* \cdot (1-\rho_{HV}^*) \cdot (1-gradV^*)]$ shows better performance for the estimation of the freezing level. We magnified the time window to assess the variation of the FL and the accuracy of the FLA using the profile $P_{26}$ over one year of radar data. This is shown in Figure 9. Figure 9 confirms the good performance of the combination $P_{26}$ on the FL estimation, even when increasing

the time window, with errors close to 200 m and high correlation coefficients. As expected, the larger the time window, the





larger the errors, as the number of profiles included in the analysis increases and the measured FL is likely to vary during this period.

Examples of the estimation of the FL on stratiform and convective events for profiles $P_{26}$ and $P_{16}$ are shown in Figure 10. Figures 13a and 13b, displayed over an HTI plot of reflectivity, show that running the algorithm using the combination $P_{26}$ in

both, stratiform or convective events, yields a good performance of the FL estimation, especially for stratiform events, where the FL height and the rain zone is accurately defined. For the convective event, the FL is correctly identified, albeit the bottom of the ML is not detected. This highlights the problem of the algorithm when describing low-altitude melting layers based on VPs. Similarly, Figures 10c and 10d, displayed over an HTI plot of velocity gradients, exemplify the performance of the algorithm based on the profile $P_{16}$ where the variable $[1 - gradV^*]$ confirms its capacity to detect the FL.

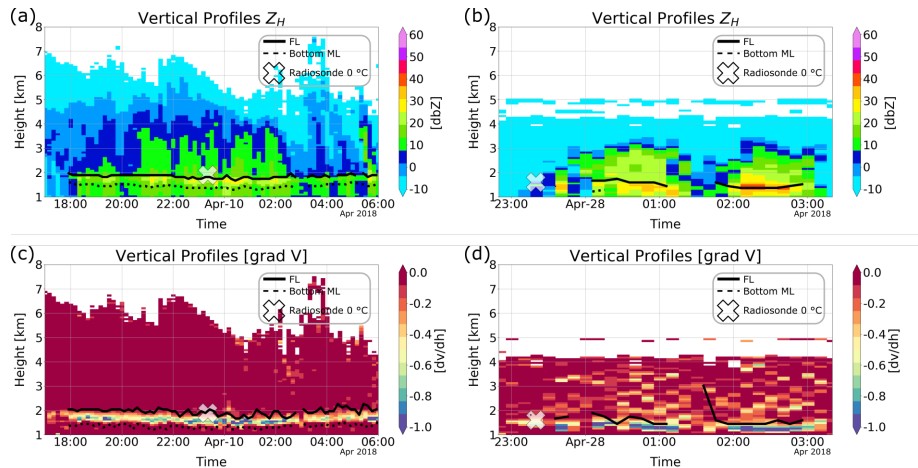

**Figure 10.** Comparison of the FLA outputs at $90°$ elevation angle using two different variables: (a) and (b) show the performance of the variable $P_{26}$ for a stratiform (top left) and a convective event (top right); (c) and (d) display the performance of the variable $P_{16}$ for a stratiform (bottom left) and a convective event (bottom right).

## 5.4   FL estimation from QVP

The FLA is applied to QVPs generated from scans at three different elevation angles ($4°$, $6°$ and $9°$). After several trials on the parameters $k$ and $w$ of the algorithm, only the highest elevation produced satisfactory results on the FL estimation. The explanation of this has its foundation in Figure 3, where QVPs from lower elevation angles display shapes that make the implementation of the algorithm difficult. For instance, the profile of $\rho_{HV}$ exhibits a peak related to the ML but above this peak

the values of $\rho_{HV}$ decrease sharply, whilst the profile of $Z_H$ exhibits smoother peaks and when the normalisation process is carried out, the parameter $k$ is not able to identify gradients strong enough to be related to the ML. Thus, after several trials and supported by the analysis presented in Section 5.1, we decided not to use the lower elevation angles ($4°$ and $6°$). Using the same windows as in the vertical profiles, we computed several performance metrics ($r, MAE, RMSE$) between the $0°C$ isotherms



measured by the radiosondes and the estimated by the FLA. The performance of the algorithm using different profiles and a
time window of 60-min (i.e. using radar profiles 30-min before and after the radiosonde timestamp) is shown in Figure 11.

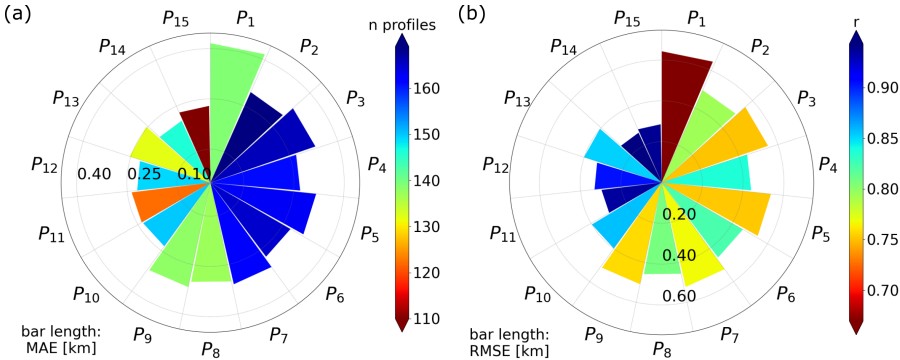

**Figure 11.** Errors in the FL estimation for QVPs using a $\pm 30$ minute window. In (a) the bar length represents the MAE (in km) for every polarimetric combination and colour represents the number of QVPs analysed for every polarimetric combination; in (b) the bar length represents the RMSE (in km) and colour represents r.

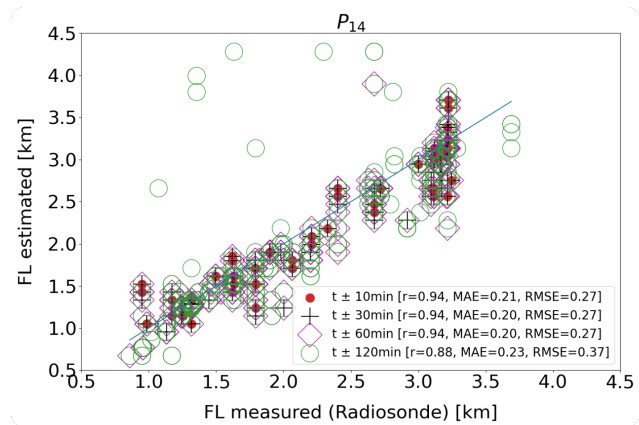

**Figure 12.** Heights of the $0°C$ isotherm measured by radiosonde versus FL estimated by the algorithm for several time windows using QVPs from $9°$ elevation scans. The 1:1 line is shown in blue. MAE and RMSE in km.

Figure 11a shows that the number of profiles covered by the time window is somewhat more significant than the number of profiles covered in the implementation of the VPs. This is expected given that the coverage area of the PPIs from where the QVPs were constructed is greater compared to the vertical scans. Overall, the four indicators in Figure 11 stress the influence of $Z_H$ and $\rho_{HV}$ in the estimation of the FL height and reveal that appending the combination $Z_{DR}^*$ to the analysis, i.e. $P_{12}$,
$P_{14}$ or $P_{15}$ improve the delimitation of the ML, given that these combinations exhibit high values of correlation $((r))$ and the errors are below 300 m. Based on these results, and combined with a visual assessment of the outputs of the algorithm over a





whole year of precipitation profiles, we concluded that the profile that combines $Z_H^*$, $Z_{DR}^*$ and $(1 - \rho_{HV}^*)$, i.e. $P_{14}$ provides a better estimation of the freezing level. The performance of the algorithm using this combination is shown in Figure 12.

Figure 12 shows that error and correlation coefficient decrease as the time-interval increase. Given that the errors are close

to 250 m, this combination proves to be accurate for the FL estimation, making allowance for the original resolution of the scans (600 m). Two examples of the outputs of the algorithm selecting the combinations $P_{10}$ and $P_{14}$ as the final product are shown in Figure 13 for the same stratiform and convective events as in section 5.3. Figures 13a and 13b show the outputs of the algorithm using the combination $P_{10}$, displayed on a base-HTI plot of $Z_H$, where the performance for a stratiform event is acceptable, given the presence of random, noticeable peaks along to the duration of the event, whereas the performance of

this combination for the shower event is less accurate, due to the absence of strong gradients on the profile computed in the first part of the algorithm and emphasised on the second one. On the other hand, the combination $P_{14}$, displayed on a base-HTI plot of $Z_{DR}$, shows better performance, especially for the stratiform event shown in Figure 13c, as the FL is precisely detected and the delineation of the rain region is well-executed. For the convective event of Figure 13d, the outputs of the algorithm are accurate for the FL estimation although some gaps are present due to the filtering of profiles in the first part of the algorithm.

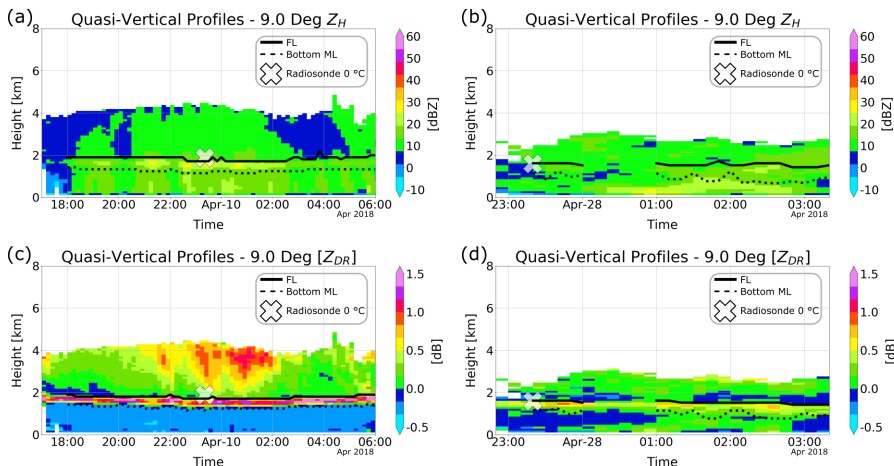

**Figure 13.** Comparison of the FLA outputs at $9°$ elevation angle using two different variables: (a) and (b) show the performance of the variable $P_{10}$ for a stratiform (top left) and a convective event (top right); (c) and (d) display the performance of the variable $P_{14}$ for a stratiform (bottom left) and a convective event (bottom right).

# 6  Discussion

We constructed VPs and QVPs of polarimetric variables to explore precipitation events and its features. As shown in Figure 2, both types of profiles display differences that are influenced by the scan elevation angle and the methods used for the construction of the profiles (Giangrande et al., 2008; Kumjian and Lombardo, 2017; Ryzhkov et al., 2016).

From the analysis of the QVPs and VPs, we observed that $Z_H$ is the variable that is most susceptible to the different types of





precipitation on both types of profiles, allowing the characterisation of the rain profiles, as previously explored by Fabry and Zawadzki (1995), Kitchen et al. (1994) and Klaassen (1988). Nevertheless, this also accentuates the trouble on the detection of the FL based only on the reflectivity profiles, although some authors, e.g. Gourley and Calvert (2003) developed methods based mainly on this variable. This emphasises the need to incorporate other polarimetric variables to the analysis.

Regarding $Z_{DR}$, this variable raises several questions about its potential to estimate the FL. $Z_{DR}$ is a polarimetric variable

prone to calibration errors (Vivekanandan et al., 2003), and our datasets are not the exception, as shown in Figure 2c. We decided not to carry out a calibration process at this point, because knowledge of the FL height is necessary, as suggested by Gorgucci et al. (1999), Gourley et al. (2009) or Park et al. (2005). Moreover, the values of $Z_{DR}$ greatly vary regarding the elevation angle, as shown in Figures 2c and 2d and proved by Ryzhkov et al. (2005), where they found that $Z_{DR}$ decrease with elevation angles for weather targets. Regardless of these drawbacks, the profiles of $Z_{DR}$ show its sensitivity to the variance on

the hydrometeor characteristics as can be seen in Figure 3b, enabling the estimation of the FL. This was explored by Brandes and Ikeda (2004), Giangrande et al. (2008), Matrosov et al. (2007), Wolfensberger et al. (2016) and although some of them decided not to include this variable into their algorithms, we included it into the analysis to explore its effectiveness for the FL estimation.

On the other hand, $\rho_{HV}$ stands out as a tell-tale of the ML, on both QVPs and VPs, as shown in Figures 2e, 2f and 3c. This

agrees with the findings of Brandes and Ikeda (2004), Matrosov et al. (2007), Shusse et al. (2011) and Tabary et al. (2006), that based their algorithms partially or solely on this polarimetric variable. Also, we analysed the quality of the radar datasets on several convective and stratiform events based on this variable and found that $\rho_{HV}$ in the rain medium is around $0.99$ and so the quality of this variable is reliable for further processing.

For our datasets, $\Phi_{DP}$ stands out as the profile where the phase of the hydrometeors yields in complex signatures that are

difficult to classify, as shown in Figure 3d. Since the elevation angles used for the construction of the QVPs are below $10°$, the peaks in $\Phi_{DP}$ related to the ML are weak and not well defined, although when using higher elevation angles, the peak in $\Phi_{DP}$ should increase, as shown by Trömel et al. (2014). There are also other peaks present at the top of the QVPs, but these peaks are likely related to the dendritic growth layer (Kaltenboeck and Ryzhkov, 2017). Additionally, the VPs of $\Phi_{DP}$ presented in Figure 3d differs to the profiles showed by Brandes and Ikeda (2004) as in their Figure 1, $\Phi_{DP}$ increase on the ML, but for our

VPs, there is an inverse peak caused by the ML.

Finally, the fall velocity of hydrometeors has been widely used on several algorithms for the ML characterisation (Baldini and Gorgucci, 2006; Vignal et al., 1999; White et al., 2002). In most of the cases, the velocity profiles are used in combination with other variables (reflectivity, wind profilers data and Doppler velocity width) to estimate the height of the FL, but these data are not available on the UKMO weather radar network. As can be seen in Figure 2i, the Doppler velocity profiles displayed

in the height-versus-time format are a great tool to observe the development of the precipitation events as they describe the increase of the fall velocity of hydrometeors as the snowflakes melt into raindrops. Also, there is an area where the velocity is nearly zero, describing the shift between ice, snow and melting particles. Hence, to take advantage of these measurements, we propose a simple but effective way to incorporate this variable into the FL estimation, computing the derivative of the profile (as described in Section 2), that transforms the profile into a similar format of the polarimetric variables that enable the addition





of this profile into an automated peak-detection algorithm, as can be seen in Figures 2j and 3e (dotted line).

These signatures on the polarimetric profiles and in combination with the radiosonde datasets helped us to corroborate that the top of the ML is related to the $0°C$ isotherm (FL) and we also observed that the bottom of the ML is related to the beginning of the rain region, especially in stratiform precipitation as can be seen in Figures 2 and 3 and demonstrated by previous studies (Brandes and Ikeda, 2004; Kumjian and Lombardo, 2017; Rico-Ramirez et al., 2007; Ryzhkov et al., 2016).

Based on all the different signatures observed in the ML, we designed an algorithm that detects the strong gradients in the profiles caused by the ML to estimate the FL. The algorithm is based on the method proposed by Wolfensberger et al. (2016), where the combination of polarimetric profiles enables the detection of the FL. We modified their algorithm to include all combinations of polarimetric variables to evaluate the capability of each variable and improve the accuracy of the FL estimation. We proposed a method that considers the ML and its boundaries as peaks and valleys within a profile and then enhancing them

enables accurate detection of the FL. This is a simple method that differs from previous studies where the ML and its boundaries are detected by complex methods that compute second-order statistics of polarimetric profiles (Baldini and Gorgucci, 2006), assume idealised profiles (Brandes and Ikeda, 2004), use a curvature detection method (Fabry and Zawadzki, 1995) or methods that rotate the coordinate system to locate the ML boundaries (Rico-Ramirez and Cluckie, 2007). Moreover, some of these methods are tailored exclusively to a specific scanning strategy or type of profiles, so we proposed two parameters ($k$ and $w$)

intended to be calibrated depending on the type of profiles.

We also assessed the consistency between QVPs and VPs of $Z_H$ to make sure that the low-elevation angles available in our datasets are still useful to compute reliable QVPs, as Ryzhkov et al. (2016) suggested that QVPs should be built from data collected at higher elevation angles that exceed $20°$. The results for the lower elevation angles ($4°$ and $6°$) agree with the findings by Ryzhkov et al. (2016) proving that decreasing the antenna elevation degrades the resolution of the QVPs. However,

the QVPs collected at $9°$ elevation angles are still in good agreement with the VPs of $Z_H$; in overall, there is a good agreement between datasets in stratiform events as the correlation coefficient is close to 0.7, but in convective events, the differences between the profiles increase, as can be seen in Figure 6 and Figure 7. Therefore, we concluded that the QVP can be generated from elevation scans of $9°$ as the effects of beam broadening and horizontal inhomogeneity are not as pronounced as expected and this enables the use of these QVPs of polarimetric variables for the detection of the FL.

The results of the FLA were validated by comparing with the measured FL height ($FL_{RS}$) from radiosonde data. Using this data, we demonstrated the potential of each one of the polarimetric variables to estimate the FL, by presenting performance metrics of all the available variables and type of profiles, as shown in Sections 5.2 and 5.3. For the VPs, we demonstrated that the proposed profile $gradV$ is helpful for the estimation of the FL (see Figures 10c and 10d, especially in combination with other variables e.g. $P_{26} = [Z_H^* \cdot (1 - \rho_{HV}^*) \cdot (1 - gradV^*)]$ that generate accurate estimations of the FL, regardless if it

was applied to convective or stratiform events. It is worth noting that the combination $P_{10}$ is similar to the algorithm proposed by Wolfensberger et al. (2016) (although the refinement process of the estimation of the FL is different). However, in our case, it showed less accuracy, especially when the profiles were related to convective-type events, as shown in Figures 13c and 13d. Regarding the bottom of the ML, it is important to highlight that only a visual assessment enables the validation of the performance of the algorithm in this matter.





On the other hand, for the QVPs, the incorporation of $Z_{DR}$ into the algorithm showed to add valuable information to the identification and characterisation of the ML. Hence the combination $P_{14}$ is selected as the one with the best performance for the detection of the FL when applied to the QVPs. As can be seen in Figure 11, the accuracy improves in comparison to the profile $P_{10}$ that only includes the combination of $Z_H^*$ and $1 - \rho_{HV}^*$.

   Therefore, we select these two profiles $P_{14}$ and $P_{26}$ for QVPs and VPs, respectively, as the combinations that achieve the
higher accuracy on the estimation of the FL and, at a certain degree, the ML characterisation. These combinations proved to be accurate, with an average error close to the resolution of the radar and the mismatch in time and space. The proposed algorithm produces errors within 200m in the estimation of the FL estimation, which is consistent with previous work (Brandes and Ikeda, 2004; Baldini and Gorgucci, 2006; Kitchen et al., 1994; Wolfensberger et al., 2016). Finally, it is worth noting that the algorithm enables the detection of the FL based on radar measurements only without relying on FL estimations from NWP
model runs. This allows the implementation of radar rainfall correction schemes or hydrometeor classification algorithms based on radar measurements only.

## 7   Conclusions

   In this paper, we generated QVPs and VPs of polarimetric variables collected by an operational C-band radar to explore the identification of the FL. We observed that QVPs of polarimetric radar variables reveal signatures that are difficult to observe
in traditional PPI scans. Also, QVPs show more precipitation events than VPs as the VPs can only measure events that are effectively happening above the radar. Even more, for the datasets used in this work, scans taken at $90°$ elevation presents limitations when reading data on the first kilometre due to technical restrictions, this situation restrain the observation of rainfall features at relative lower heights, while the QVPs are not affected by this constraint.

   We performed a numerical comparison of the VP and QVPs of reflectivity to demonstrate the consistency of the measurements
involving the elevation angle of the scans. The analysis shows that QVPs generated using elevation angles at $9°$ exhibit good agreement with VPs ($r \sim 0.7$) while elevations below this elevation increase the discrepancy with vertical scans and are not suitable for the detection of the FL.

   We analysed the signatures of the polarimetric variables to characterise the ML because they represent a diversity of orientation, shape, size, and microphysical processes of the hydrometeors, we concluded that these features have an impact on the
shape of the polarimetric profiles and therefore improve the detection of the FL.

   We developed a robust, operational FLA that detects the signatures of the ML using polarimetric QVPs and VPs. The fundamentals of the design of the FLA are: A simple method to detect peaks and valleys within the profiles;

The combination of normalised variables;

And the incorporation of two parameters that can be calibrated depending on the characteristics and type of the profiles.
We propose a novel profile $[gradV]$ for velocities taken at vertical incidence, that proves to be a helpful variable for the FL estimation. We demonstrated the capability of all the available VPs and QVPs of radar variables for the estimation of the FL, providing individual performance metrics and analysing their performance on convective and stratiform events. For VPs, the com-



bination $P_{26}$, that use the normalised version of the reflectivity, the correlation coefficient and the gradient of the velocity, i.e. $[Z_H^* \cdot (1 - \rho_{HV}^*) \cdot (1 - gradV^*)]$ achieve accurate estimations of the FL. For QVPs, the combination $P_{14} = [Z_H^* \cdot (Z_{DR}^* \cdot (1 - \rho_{HV}^*)]$

is selected as the combination that produces better results for the FL estimation on both, stratiform and convective events. The FLA proves to be accurate as the errors $(MAE, RMSE)$ between the selected outputs of the FLA and the data collected by radiosonde are close to 200 m.

*Code availability.* The freezing level estimation algorithm described in this work and other radar data visualization tools used in this study will be available on request from the corresponding author.

*Author contributions.* Daniel Sanchez-Rivas was responsible for the algorithm development, validation of the algorithm and writing of the paper. Miguel Rico-Ramirez provided supervision of the work and contributed to the writing of the paper.

*Competing interests.* The authors declare that they have no conflict of interest

*Financial support.* This research has been supported by the the CONACyT (Grant Number: 637289) and the Engineering and Physical Sciences Research Council (EPSRC) (via Grant EP/I012222/1).

*Acknowledgements.* The authors would like to thank the UK Met Office for providing the polarimetric radar data and high-resolution ra-diosonde datasets through the British Atmospheric Data Centre (Met Office, 2007, 2013). This work was carried out using the computational facilities of the Advanced Computing Research Centre, University of Bristol
(http://www.bris.ac.uk/acrc/).



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
