# Peer review of "Detection of the melting level with polarimetric weather radar"

_Atmospheric Measurement Techniques, 2020_

## Referee Comment (RC1) · Anonymous Referee #1 · 23 Oct 2020

This paper describes a melting layer detection technique from vertical Profiles (VP) and quasi-vertical profiles (QVP) from polarimetric radar observations. Examples are given from a C-band operational weather radar in SE England. Apart from Zh, Zdr, phi_dp, and rho_hv, the technique includes mean Doppler velocity and the gradient of the vertical Doppler velocity.

The paper can be published in AMT but it needs to be written in a more coherent manner. Sentences don't follow each other in some cases, and more clarification is needed in some cases. I give some examples below:

1. At the end of Intro, insert a paragraph outlining what this paper is trying to achieve and how the paper is structured.

2. Line 95: By Doppler velocity, do they mean the mean radial component?

3. Line 115: What does 'visible signatures' mean? Can you quantify?

4. Line 128: The authors say "Based on the profiles of vertical velocity [V], we propose a new variable: [gradV].." - What about spectral width? Is this available from routine scans?

5. Figure 2: For the VP plots on the left side, the y-axis should go from 0 to 8 km to be consistent with the QVP plots. What about panel (j)? Why is the 0 to 1 km omitted?

6. Line 144: Define 'normalised' at this point.

7. Line 147: should 'estimate' be 'detect'?

8. Line 148: What does 'enhancements that the ML bring-up into the variables' mean?

9. Line 154: By 'elevation' do they mean 'altitude a.g.l'?

10. Lines 156-159: Grammar needs to be improved, and also the text is ambiguous; the sentence doesn't make much sense.

11. Lines 163: convective events are associated with different microphysical processes so ML doesn't apply.

12. Line 168: Doesn't the radar perform 'bird-bath' scans routinely?

13. Line 168: The sentence beginning 'Hence the Zdr ..' requires much more clarification.

14. Section 3 is verbose, not very technical and not well-written at all. Please rewrite. Also explain clearly why the peaks in Zh, Zdr and rho_hv are at different heights above ground level and explain the difference between BB and ML.

15. Line 237: Once again, explicitly say how the normalisation is performed.

16. Explain how equations (2) and (3) were derived. If published elsewhere, then insert

reference for the derivations.

17. Line 267: Explain/justify why the second derivative was chosen.

18. Line 293: "QVPs and VPs of Zh, as these variables measure similar properties of the raindrops" What does this mean?

19. Line 303: What does "resides on relative low values of reflectivity" mean?

20. What is the purpose of Section 5.1 if only the Z comparisons are given? It's not clear how it is relevant to the rest of the paper.

21. Regarding Fig. 9: What does 'FL estimated' represent exactly, that is in relation to the radar BB (peaks in all the variables), and the 0 deg C isotherm level?

22. What about attenuation corrections needed for Zh and Zdr? Were these applied?

These are some of the comments. Considerable amount of revision is needed

Please also note the supplement to this comment:
https://amt.copernicus.org/preprints/amt-2020-375/amt-2020-375-RC1-
supplement.pdf

---

## Referee Comment (RC2) · Anonymous Referee #2 · 26 Oct 2020

Review for AMT-2020-375

"Detection of the freezing level with polarimetric weather radar"

Daniel Sanchez-Rivas and Miguel A Rico-Ramirez

This study proposes an approach to estimate the freezing level (FL) using vertical/quasi-vertical profiles (VP/QVP) achieved from polarimetric radar observations. The proposed approach was applied to some selected events, and the estimated FLs were evaluated using radiosonde observations. Based on the evaluation results, the authors concluded that the combinations of ZH, HV, and the gradient of the velocity V, and ZH, HV, and ZDR for each VP and QVP method are the best predictors for the FL estimation. I think that the study was well-designed, and the focus and experimental

details and results of the study are clearly addressed in the manuscript. However, I have a basic question about the utility of this study for radar QPE and additional comments/suggestions for some other aspects presented in this study.

Major comments:

1. Utility of FL height The authors discuss the necessity of FL information for radarbased applications (e.g., QPE) in Introduction. In my opinion, what is really useful for radar applications is to provide a range of the melting layer (ML), not just a single value of FL height itself (as this study mostly devoted to find the FL height) because mixed (liquid-solid) precipitation is usually located below the FL height, and this is a significant challenge e.g. for rainfall and attenuation estimation. I am wondering what specific applications require the estimated FL height. I think that a bottom height of the ML presented in Figures 10 and 13 is much more useful than the FL height itself because the majority of scattering and propagation theories can be applied only to the region below this height (liquid precipitation or pure rain region).

2. QVP It is not clear if either time-averaged or instantaneous QVP is used in the proposed FL detection algorithm. I think that instantaneous QVP is not appropriate for the proposed algorithm because it could be affected by local storm structures (although it is derived from higher elevation angles) particularly for the ones near the radar. If the authors used time-averaged QVPs, they need to clarify it and define the averaging time window.

It might be helpful for readers to understand the QVP method if the authors provide a brief description on the background and procedures to retrieve QVP from radar observations, rather than just referring to Ryzhkov et al. (2016).

3. FL spatial variability I think that the proposed QVP method results in the average FL over the entire radar domain while the VP method yields limited FL to the radar site (if VP was obtained from a 90 degree elevation angle). I am wondering how the spatial variability of FL over the radar domain looks like, and the authors may compare the

**AMTD**
FL information retrieved from the NWP model with the one achieved from this study. It might be helpful to discuss this spatial variability issue in the discussion section as a limitation of this approach.

4. Error analysis Whereas the analyses presented in this study focused on finding the best predictors of the polarimetric radar observations, it is valuable to characterize the structure of errors resulted from the proposed methods. I think that it would be useful to demonstrate error distributions of each VP and QVP method (e.g., P14 and P26), rather than just reporting "the errors in the FL estimation using either VPs or QVPs ae within 250m."

Minor comments:

1. Line 10 Maybe better to remove "extremely."

2. Line 24-26 It would be interesting to compare the FL heights computed from between this study and the NWP model.

3. Line 87 Please define "UKMO."

4. Line 106 Please replace "twice daily" with "twice a day."

5. Table 1 I think that the "Location" in Table 1 represents coordinates on a certain projected coordinate system. Geographic coordinates are more common and please provide latitude and longitude of the radar site.

6. Figure 2 Please use consistent height (y-axis) and color scales for the same radar observables to enable easy comparisons between left and right panels for (a)–(h). Please also define "HTI" in the figure caption.

7. Line 144 Please clarify if QVPs shown in Figure 3 were time-averaged before they were normalized.

8. Line 181-182 How are "type of precipitation" and "phase of the hydrometeors" different?

AMTD
9. Line 278 It turned out that "magnitude (k) of Ppeak" was a threshold (e.g., parameter) for peak magnitude (Line 287). Please clarify it here.

10. Line 291-294 Something is missing. Please rewrite.

11. Line 300 Why do the authors compare VPs and QVPs? Is this comparison performed because the authors used instantaneous QVPs for FL estimation? I think that they (VP and QVP) are not necessarily consistent, and QVP should be used with timeaveraging to avoid local storm effects and capture the consistent vertical structure with VP.

12. Section 5.2 This section does not describe the result of this study and should be moved to the "Methodology" section.

13. Line 352 Please replace "better" with "best."

14. Line 358 Why P16? Both ZH and [grad V] are the elements of P26. [grad V] was used for P16–P31, not just for P16.

15. Line 359 Figures 10a and 10b instead of "Figures 13a and 13b?"

16. Line 361-362 The estimation procedure of the ML bottom was not described.

17. Line 383 Please replace "better" with "best."

18. Line 386 Why P10? P10 does not have to be mentioned here because the two factors shown in Figure 13 are also included in P14.

19. Line 404-407 I think that the ZDR calibration bias is not an issue in this study because relative ZDR values (e.g., normalized) are used to construct vertical profiles. ZH also contains the calibration issue.

---

## Referee Comment (RC3) · Anonymous Referee #3 · 26 Oct 2020

Reviewer Comments on "Detection of the freezing level with polarimetric weather radar", by Daniel Sanchez-Rivas, and Miguel A Rico-Ramirez, AMT-2020-375.

Overview Comments:

The manuscript describes a technique to estimate the Freezing Level (FL) height, motivated by its practical use in downstream hydrological applications. The methods blend QVPs/VPs ideas as a convenient way to summarize the dual-polarization, vertical properties to inform this retrieval.

Overall, the manuscript accomplishes the application it sets out to perform. However, the effort seems limited in that it amends previous ideas with potentially questionable inputs. Such applications may be publishable within the scope of AMT, but this seems

to require substantial edits (not a trivial re-write). The manuscript is long, yet not particularly organized in how it presents concepts, physical discussions. Most statements probably should be more conservative. It is not always clear what is original, or why this advancement matters? One radar advantage (somewhat lost with QVPs) is ability to capture FL variability spatially when compared to model output, surface, or radiosonde information. The manuscript claims originality from QVPs, but avoids when it is appropriate to use QVPs, e.g., important trade-offs for this decision. QVPs could be a smart substitution, but this choice encourages compensating errors. It is also not clear the outcome (i.e., FL estimates to match 0C Temperature) is the best target (i.e., 0C Wet Bulb Temperature).

Major comments:

1) What accuracy does one require "FL" estimates, and is this important? What is the 'value-added', aka, why this specific approach? What is the advantage over existing ML ideas?

2) QVPs are spatial averages that favor widespread precipitation, homogeneous fields. QVPs are clever, convenient, but one 'practical' issue is related to their generation – e.g., this requires more statements on the tolerance for 'when' (under what conditions) these are generated, e.g., 'how frequently' does this result in useful retrievals? What about 'edge case' QVPs that may be generated, but require filtering? Basically, how confident are we that all conditions that allow a QVP are also equally viable as inputs?

3) QVP averaging removes ability to define regions of mixed precipitation (azimuthally) as one example issue common to FL/ML literature. The variability of the FL can be substantial, studies suggesting O[500 meters] – variability as large as the melting layer – and radar sectors where 'rain' switches to 'snow'. This argues QVPs are not suitably fine-grained, would struggle in locations where this is a concern, e.g., Boodoo et al. (2010).

4) Melting onset is not at 0C temperature, rather at the 0C Wet Bulb temperature. When

viewing from radar, the height when one claims a melting response is typically lower, e.g., delay in measurement sensitivity to melting, but also the RH is often not 100%. A concern is if one becomes too interested in a retrieved 'match' to a radiosonde target that is not always correct. While the 0C temperature is the historical 'freezing level' definition, it is not the one (or only) hydrological applications care about associated with 'contaminated' radar signatures (e.g., 'bright band' shape also starts above with aggregation processes). This is partially why I suspect VP/velocity profiles are not as seeming useful in the offered, e.g., this is more a case of poor target/definitions than velocity not being a highly useful input.

5) QVP-based dual-pol measurement profiles have different issues for interpretation; For example, ZDR profiles do not rapidly increase until onset of melting (0C Wet Bulb), whereas Z is increasing above the ML owing to aggregation. Unfortunately, where these signatures occur in altitude is complicated further when aggregation, melting are not the same spatially, then averaged in a QVP. The QVP issues are exacerbated when coupled with nonuniform beam-filling NBF issues that smear profiles. This calls into question concepts for 'combining' Z and RHOHV (or variants therein) – as in Wolfensberger et al. reference – as confusing when based on QVPs. It is not clear the order of operations, and how/when one averages, combines such fields. It makes a difference in the eventual input profile validity. Moreover, it may lead to solutions that 'work', but come to a matching answer for the wrong reasons.

6) Effort is spent on explaining dual-polarization signatures (QVP), but the important process aspects are not too well described. Radar response to processes/properties (signatures) to include melting, aggregation, break-up/fallout, etc., are undoubtedly difficult. These processes and observed properties are smoothed/complicated further in response to known radar (system) bias, NBF, etc. Averaging and other processing details distort things further, esp. regions that preferentially feature different density or mass flux into these melting layers, RH, vertical motions, etc. There are a few resources for discussion on QVP signatures of the bright band (and reasons for its

variability), e.g., Kumjian et al. (2016).

I call attention to nonuniform beam filling NBF in particular, and melting onset expectations. Illustrations for potential offsets in radar quantities are found in Ryzhkov (2007). For intermediate tilts being used for QVPs, the expectation should be for modest biases in quantities owing to NBF (e.g., smearing) – This is complicated by the QVP averaging if the fields are not homogeneous. It is possible to model how well certain combinations of quantities may demonstrate compensating issues if one attempted to multiply those profiles, at different tilts, etc. – much of that would also arguably change on when those QVP averaging was performed (multiply before QVP, or QVP before multiply?). Again, it does not make sense (to this reviewer) how Z and RHOHV fields can be multiplied to generate a useful profile without factoring in several details (thus, also not surprised it may not apply consistently well, either).

Minor Comments:

1) The 'freezing level' is a poor term, persists in operations. Perhaps 'melting level', as frozen media begins to melt at that level.

2) The authors use examples for the QVP, VP profiles in several figures. Critically, I find these examples often physically nonintuitive, even when the authors imply these as only meant as examples. For example, one expects the Z peak to be higher in altitude (above) of the ZDR peak, with the ZDR and RHOHV peaks located at similar altitudes. If the authors retain the physical discussions on the dual-polarization signatures, the reasons for such relative behaviors are perhaps more important. These are also far less commonly described.

3) I had an impression velocity gradient ideas were being presented as novel/unique. The authors should likely consult the profiling radar literature (e.g., works of C. Williams, other profiling radar echo classification manuscripts) that commonly use gradients of mean Doppler velocity in their efforts. As above, I suspect velocity is more accurate / informative profile input (when available) for assessing the wet bulb zero for reasons of

its improved vertical resolution and sensitivity to its relative 'change point' with melting onset. I suspect open-code/python change point / inflection techniques would also apply vertically as compared to gradient ideas, too.

---

## Author Comment (AC1) · 1 Jan 2021

1. At the end of Intro, insert a paragraph outlining what this paper is trying to achieve and how the paper is structured

   (a) *We'll add a paragraph as: "The main objective of this work is to present an automated, operational and robust algorithm that can accurately estimate the FL based on the combination of QVPs or VPs of polarimetric measurements collected from operational dual-polarisation weather radars. The algorithm outputs are validated using FL heights from high-resolution radiosonde data. Note that the proposed algorithm is not intended to replace NWP-based FL estimation methods, but it is an alternative way to estimate the FL when only polarimetric weather radar measurements are available. The article is organized as follows: the next section will describe the datasets used for the design and validation of the algorithm. The aim of Section 3 is to examine the footprints of the melting layer*

on both QVPs and VPs of polarimetric variables. Section 4 provides a detailed explanation of the design of the algorithm. Results, implementation, validation and several examples of the outputs of the algorithm are presented in Section 5. Section 6 provides a discussion on the performance and implementation of the algorithm. Finally, Section 7 provides a summary of the conclusions from this work."

2. Line 95: By Doppler velocity, do they mean the mean radial component?

   (a) *The reviewer is right: the sentence needs further explanation; we'll add a phrase as: "Mean radial velocity ($V$) measurements of the observed droplets is available"*

3. Line 115: What does 'visible signatures' mean? Can you quantify?

   (a) *We agree the sentence needs further explanation. We'll rephrase as follows: "A total of 94 rainfall events with visible signatures of the ML on $Z_H$ or $\rho_{HV}$ were selected, i.e. an enhancement up to 30 dBZ on $Z_H$ or $\rho_{HV}$ constantly decreasing below 0.90. Also, from the total events, only 25 rain events observed by the radar shown a suitable temporal matching with the data collected by the radiosondes, i.e. the difference in time between measurements do not exceed 3 hours."*

4. Line 128: The authors say "Based on the profiles of vertical velocity [V], we propose a new variable: [gradV].." – What about spectral width? Is this available from routine scans?

   (a) *The Spectral width variable was not available in the analysed radar datasets.*

5. Figure 2: For the VP plots on the left side, the y-axis should go from 0 to 8 km to be consistent with the QVP plots. What about panel (j)? Why is the 0 to 1 km omitted?

   (a) *As described on line 125, data collected at vertical incidence is contaminated by spurious echoes. Still, we'll modify the plot so the y-axis is consistent on both sides, enabling a straightforward comparison.*

6. Line 144: Define 'normalised' at this point.

   (a) *We'll rephrase as follows: "For comparison purposes, the VPs and QVPs of polarimetric variables are normalised (scaling each feature into the range [0, 1]) to intensify the features observed in the ML; each variable maximum is set to 1, whereas the minima are set to 0. Examples of normalised QVPs and VPs related to a stratiform event*

*are shown in Figure 3 along with the closest-in-time radiosonde data where the temperature reaches 0 °C ($FL_{RS}$)."*

7. Line 147: should 'estimate' be 'detect'?

    (a) *Agreed and corrected*

8. Line 148: What does 'enhancements that the ML bring-up into the variables' mean?

    (a) *We'll rephrase as follows: "Given that the main objective of this work is to detect the FL based on the geometric features of the polarimetric profiles, herein, we will try to explain the ML signatures and how it shapes the structure of the profiles."*

9. Line 154: By 'elevation' do they mean 'altitude a.g.l'?

    (a) *Indeed. This will be corrected in the revised version.*

10. Lines 156-159: Grammar needs to be improved, and also the text is ambiguous; the sentence doesn't make much sense.

    (a) *We'll rephrase as: "The reflectivity ($Z_H$) is represents the power backscattered by precipitation particles, thus providing information about the concentration, size, phase and water content of the hydrometeors (Gourley and Hong, 2014). In figures 2a and 2b it can be seen that the values of $Z_H$ on both QVPs and VPs show similar intensities. These aspects will be analysed on the following sections."*

11. Lines 163: convective events are associated with different microphysical processes so ML doesn't apply.

    (a) *Agreed, we'll rephrase as follows: "Whilst for convective events, the profiles of $Z_H$ do not show the BB feature, therefore the estimation of FL based only on this variable and for this type of events is not feasible."*

12. Line 168: Doesn't the radar perform 'bird-bath' scans routinely?

    (a) *Yes, the bird-bath scans are the VPs. Please note that in lines 169-170 we explained the need to know first the height of the FL to apply a bias-correction to $Z_{DR}$.*

13. Line 168: The sentence beginning 'Hence the Zdr ..' requires much more clarification.

(a) *We agree the sentence needs further explanation. We'll rephrase this sentence as follows: "From Figure 2c we can observe that $Z_{DR}$ is not calibrated, as we expect near-to-zero values for $Z_{DR}$ in rain region for vertically pointing measurements as raindrops are symmetrical on average when observed from underneath (Gorgucci et al., 1999). Non-zero $Z_{DR}$ values in rain are a strong indicator of uncalibrated $Z_{DR}$ measurements, and a subsequent analysis of 'birdbath' scans in light rain confirmed a negative offset. Hence, $Z_{DR}$ measurements must be corrected if $Z_{DR}$ is intended for radar QPE; this reaffirms the importance of the detection of the freezing level, as it helps to set the upper height for the implementation of a $Z_{DR}$ calibration algorithm."*

14. Section 3 is verbose, not very technical and not well-written at all. Please rewrite. Also explain clearly why the peaks in Zh, Zdr and rho_hv are at different heights above ground level and explain the difference between BB and ML.

   (a) *We will revise this section to improve its readability.*

15. Line 237: Once again, explicitly say how the normalisation is performed.

   (a) *We'll rephrase as: "These two profiles are normalised and combined into a single profile ($P_{comb}$) as suggested by Wolfensberger et al. (2016), but using different thresholds that are related to drizzle, heavy rain, snow and ice (Kumjian, 2013; Fabry, 2015). The normalisation is carried out using the min-max normalisation procedure:*

$$X_n = \frac{X - min(X)}{max(X) - min(X)} \tag{1}$$

   *where $X$ is the original value and $X_n$ is the normalized value. Here, the values of $Z_H$ between 5 and 60 dBZ are normalised between 0 and 1, respectively: $[Z_H(dBZ)[5, 60] \rightarrow Z_H^*[0, 1]]$, whereas the values of $\rho_{HV}$ between 0.85 and 1 are normalised between 0 and 1: $[\rho_{HV}( )[0.85, 1] \rightarrow \rho_{HV}^*[0, 1]]$. Values outside these intervals are fixed to 0 and 1, correspondingly. Note that (\*) in the polarimetric variable indicates a normalised variable."*

16. Explain how equations (2) and (3) were derived. If published elsewhere, then insert reference for the derivations.

   (a) *We'll rephrase as follows: "Once $U_L$ is defined, the algorithm identifies the $0\,°C$ height based on a profile that is the result of the combination of several polarimetric profiles as follows.*
   *The profiles of the available polarimetric variables have an upper limit set by $U_L$ to search for the FL. Every individual profile is normalised. Note that $Z_H^*$ and $\rho_{HV}^*$ were already normalised in step*

*1.a, whereas the rest of the variables are normalised using the min-imum and maximum values of each variable. The variables were normalised based on the QVP/VP patterns observed in Section 3, i.e. variables where the peak related to the ML is orientated to the right, e.g. $Z_H$ or $Z_{DR}$ are normalised using the measured values. In contrast, variables where the ML cause a depression on the profile, are normalised using the complement of the variable, e.g. $gradV \rightarrow (1 - gradV)$. This is made to generate profiles with analogue ML peaks that enhance the footprints of the ML when combined. A new profile is computed following Equation 2 for VPs or Equation 3 for QVPs:"*

17. Line 267: Explain/justify why the second derivative was chosen.

    (a) *Please note that this is discussed in lines 284-287 and illustrated in Figure 5.*

18. Line 293: "QVPs and VPs of Zh, as these variables measure similar properties of the raindrops" What does this mean?

    (a) *We'll rephrase as: "Both VPs and QVPs proved to be an efficient way to monitor the temporal evolution of the ML. But the elevation angle from where the QVPs were taken affects in different ways to each variable, as described in Section 3 and shown in Figures 2 and 3. As $Z_H$ is the variable less prone to significant variations due to the elevation angle, we analysed the consistency between the $Z_H$ profiles constructed from different elevation angles. To some extent, the consistency between these profiles increases the confidence of the QVPs utilisation as an input of the algorithm. For the rest of the variables is not possible to quantify the consistency as they represent different properties of the hydrometeors, depending on the elevation angle from they were taken."*

19. Line 303: What does "resides on relative low values of reflectivity" mean?

    (a) *We'll rephrase as: "Figure 6a shows that lower values of reflectivity are similarly depicted on both, VPs and QVPs. These values are related to light and moderate rain rates, expected on stratiform-type events."*

20. What is the purpose of Section 5.1 if only the Z comparisons are given? It's not clear how it is relevant to the rest of the paper.

    (a) *This section is intended as a validation of the reflectivity QVPs. We believe it is necessary to assess the consistency between VPs and QVPs as the FL algorithm is based on the geometry of the profiles. But apart from $Z_H$, it is not possible to compare the figures of the*

*other polarimetric variables due to the azimuthal averaging on the construction of the QVPs.*

21. Regarding Fig. 9: What does 'FL estimated' represent exactly, that is in relation to the radar BB (peaks in all the variables), and the 0 deg C isotherm level?

    (a) *At this point, we compared the FL height measured by the radiosonde and the output of the algorithm (i.e. the BB top in the enhanced profile) described in step 2.d (lines 274-276).*

22. What about attenuation corrections needed for Zh and Zdr? Were these applied?

    (a) *We are aware that the attenuation is an error source for radar QPE. But the height of the FL is essential to implement attenuation correction algorithms as they can only be applied in the rain region. For the events analysed in this paper, we consider that the signal attenuation is negligible as we are using high-elevation scans to construct the QVPs and set a relatively short-range as a constraint for the algorithm's implementation. A glance to the generated $\Phi_{DP}$ profiles reveals that larger accumulations of $\Phi_{DP}$ are most likely related to the DGZ rather than to heavy rain.*

**References**

Fabry, F. (2015). *Radar Meteorology: Principles and Practice*. Cambridge University Press.

Gorgucci, E., Scarchilli, G., and Chandrasekar, V. (1999). A procedure to calibrate multiparameter weather radar using properties of the rain medium. *IEEE Transactions on Geoscience and Remote Sensing*, 37(1 PART 1):269–276.

Gourley, J. J. and Hong, Y. (2014). *Radar hydrology: principles, models, and applications*. CRC Press.

Kumjian, M. (2013). Principles and applications of dual-polarization weather radar. Part I: Description of the polarimetric radar variables. *Journal of Operational Meteorology*, 1(19):226–242.

Wolfensberger, D., Scipion, D., and Berne, A. (2016). Detection and characterization of the melting layer based on polarimetric radar scans. *Quarterly Journal of the Royal Meteorological Society*, 142(2000):108–124.

---

## Author Comment (AC2) · 1 Jan 2021

**Major comments:**

1. Utility of FL height. The authors discuss the necessity of FL information for radar-based applications (e.g., QPE) in Introduction. In my opinion, what is really useful for radar applications is to provide a range of the melting layer (ML), not just a single value of FL height itself (as this study mostly devoted to find the FL height) because mixed (liquid-solid) precipitation is usually located below the FL height, and this is a significant challenge e.g. for rainfall and attenuation estimation. I am wondering what specific applications require the estimated FL height. I think that a bottom height of the ML presented in Figures 10 and 13 is much more useful than the FL height itself because the majority of scattering and

propagation theories can be applied only to the region below this height (liquid precipitation or pure rain region).

    (a) *We completely agree with the reviewer on the importance of accurate detection of the bottom of the Melting Layer as most of the QPE algorithms can only be applied in the rain region. Unfortunately, if the output of the algorithm is the bottom of the ML, it would be challenging to validate it using the radiosonde datasets or some other instrument. Hence, the proposed algorithm detects both the FL and the bottom of the ML based on the geometry of the profiles and the FL is validated using radiosonde data. Then, the ML bottom can be determined using a fixed ML thickness or by using the output of the algorithm.*

2. QVP. It is not clear if either time-averaged or instantaneous QVP is used in the proposed FL detection algorithm. I think that instantaneous QVP is not appropriate for the proposed algorithm because it could be affected by local storm structures (although it is derived from higher elevation angles) particularly for the ones near the radar. If the authors used time-averaged QVPs, they need to clarify it and define the averaging time window. It might be helpful for readers to understand the QVP method if the authors provide a brief description on the background and procedures to retrieve QVP from radar observations, rather than just referring to Ryzhkov et al. (2016).

    (a) *We are aware of the advantages of using time-averaged QVPs, and we did some tests using time-averaged QVPs. The algorithm considers this situation with the parameter k, which is helpful to deal with the smoothness caused by the time-averaging of the profiles, e.g. in figure 2, the profiles are averaged using a time-window of 30 minutes, and the parameter k is modified to allow lower values on the resulting profiles. The estimated FL do not vary that much. Hence, we decided to display examples in the instantaneous QVPs format at this is the most common format of QVPs. We'll expand this in the discussion section.*

3. FL spatial variability. I think that the proposed QVP method results in the average FL over the entire radar domain while the VP method yields limited FL to the radar site (if VP was obtained from a 90 degree elevation angle). I am wondering how the spatial variability of FL over the radar domain looks like, and the authors may compare the FL information retrieved from the NWP model with the one achieved from this study. It might be helpful to discuss this spatial variability issue in the discussion section as a limitation of this approach.

    (a) *A strong motivation for this work was to avoid relying on NWP products. One of the advantages of the algorithm is that it enables the*

[Figure]

Figure 1: Instantaneous QVPs and FLe, related to a stratiform–type rain event.

[Figure]

Figure 2: Time–averaged QVPs and FLe, related to a stratiform–type rain event.

*estimation of the FL based entirely on the radar data; this is really helpful to implement corrections that depend on hydrometeor discrimination. We agree with the reviewer that there is a spatial variability of the FL over the radar domain. Still, after weighing the options, we considered that for the FL accuracy required in radar corrections, a straightforward algorithm and its validation using radiosonde surpass the complexity of data retrieved from numerical models and its computationally expensive runs, as showed by Hall et al. (2015) or Mittermaier and Illingworth (2003) . We'll discuss the spatial variability of the FL and the limitation of the algorithm in section 5.*

4. Error analysis. Whereas the analyses presented in this study focused on finding the best predictors of the polarimetric radar observations, it is valuable to characterize the structure of errors resulted from the proposed methods. I think that it would be useful to demonstrate error distributions of each VP and QVP method (e.g., P14 and P26), rather than just reporting "the errors in the FL estimation using either VPs or QVPs are within 250m."

(a) *We'll add an error analysis in section 5, comparing the detected FL depending if QVPs or VPs were used as input of the algorithm.*

**Minor comments:**

1. Line 10 Maybe better to remove "extremely."

    (a) *Noted*

2. Line 24–26 It would be interesting to compare the FL heights computed from between this study and the NWP model.

    (a) *Please refer to the answer of major comment No. 3.*

3. Line 87 Please define "UKMO."

    (a) *Corrected, UKMO refers to the UK Met Office.*

4. Line 106 Please replace "twice daily" with "twice a day."

    (a) *Corrected*

5. Table 1 I think that the "Location" in Table 1 represents coordinates on a certain projected coordinate system. Geographic coordinates are more common and please provide latitude and longitude of the radar site.

    (a) *Corrected*

6. Figure 2 Please use consistent height (y–axis) and color scales for the same radar observables to enable easy comparisons between left and right panels for (a)–(h). Please also define "HTI" in the figure caption.

    (a) *As described on line 125, data collected at vertical incidence is contaminated by spurious echoes. Still, we'll modify the plot so the y-axis is consistent on both sides, enabling a straightforward comparison.*

7. Line 144 Please clarify if QVPs shown in Figure 3 were time-averaged before they were normalized.

    (a) *Please check the answer provided above*

8. Line 181–182 How are "type of precipitation" and "phase of the hydrometeors" different?

    (a) *We agree with the reviewer, we'll rephrase as: "The correlation coefficient ($\rho_{HV}$) measures the correlation between $Z_H$ and $Z_V$ measurements and it is sensitive to the distribution of particle sizes and shapes, hence being sensitive the phase of the hydrometeors, becoming a valuable hydrometeor classifier helping to identify non-meteorological echoes (Islam and Rico-Ramirez, 2014). "*

9. Line 278 It turned out that "magnitude (k) of Ppeak" was a threshold (e.g., parameter) for peak magnitude (Line 287). Please clarify it here.

   (a) *We'll rephrase as: "An adequate choice of the magnitude of the parameter ($k$) is important to discard profiles with a $P_{peak}$ that is not strong enough to be related to the ML.*

10. Line 291–294 Something is missing. Please rewrite.

    (a) *We'll rephrase as: "Both VPs and QVPs proved to be an efficient way to monitor the temporal evolution of the ML. But the elevation angle from where the QVPs were taken affects in different ways to each variable, as described in Section 3 and shown in Figures 2 and 3. As $Z_H$ is the variable less prone to significant variations due to the elevation angle, we analysed the consistency between the $Z_H$ profiles constructed from different elevation angles. To some extent, the consistency between these profiles increases the confidence of the QVPs utilisation as an input of the algorithm. For the rest of the variables is not possible to quantify the consistency as they represent different properties of the hydrometeors, depending on the elevation angle from they were taken."*

11. Line 300 Why do the authors compare VPs and QVPs? Is this comparison performed because the authors used instantaneous QVPs for FL estimation? I think that they (VP and QVP) are not necessarily consistent, and QVP should be used with timeaveraging to avoid local storm effects and capture the consistent vertical structure with VP.

    (a) *This section is somewhat intended as a validation of the construction of the QVPs. We consider it necessary to assess the consistency between both types of representations as the FLe algorithm is based on the geometry of the profiles. But apart from $Z_H$, it is not possible to compare the figures of the polarimetric variables due to the azimuthal averaging on the construction of the QVPs.*

12. Section 5.2 This section does not describe the result of this study and should be moved to the "Methodology" section.

    (a) *We agree with the reviewer, we'll modify this section.*

13. Line 352 Please replace "better" with "best."

    (a) *Corrected*

14. Line 358 Why P16? Both ZH and [grad V] are the elements of P26. [grad V] was used for P16–P31, not just for P16.

(a) *The purpose of showing the performance of $P_{16}$ in Figures 10c and 10d is to emphasise the value of the proposed variable $gradV$. If the reviewer consider that this part is not necessary or repetitive, we are willing to leave it out of the manuscript.*

15. Line 359 Figures 10a and 10b instead of "Figures 13a and 13b?"

    (a) *Corrected*

16. Line 361-362 The estimation procedure of the ML bottom was not described.

    (a) *Due the importance of the ML bottom, we'll elaborate on this in section 3.*

17. Line 383 Please replace "better" with "best."

    (a) *Noted*

18. Line 386 Why P10? P10 does not have to be mentioned here because the two factors shown in Figure 13 are also included in P14.

    (a) *We use the variable $P_{10}$ to compare the different outputs of the algorithm, but we are willing to remove it if the reviewer considers it necessary.*

19. Line 404-407 I think that the ZDR calibration bias is not an issue in this study because relative ZDR values (e.g., normalized) are used to construct vertical profiles. ZH also contains the calibration issue.

    (a) *We agree with the reviewer, but we want to emphasize the necessity of the knowledge of the FL before the implementation of the $Z_{DR}$ calibration procedure.*

**References**

Hall, W., Rico-Ramirez, M. A., and Krämer, S. (2015). Classification and correction of the bright band using an operational C-band polarimetric radar. *Journal of Hydrology*, 531:248–258.

Islam, T. and Rico-Ramirez, M. A. (2014). An overview of the remote sensing of precipitation with polarimetric radar. *Progress in Physical Geography*, 38(1):55–78.

Mittermaier, M. P. and Illingworth, A. J. (2003). Comparison of model-derived and radar-observed freezing-level heights: Implications for vertical reflectivity profile-correction schemes. *Quarterly Journal of the Royal Meteorological Society*, 129(587 PART A):83–95.

---

## Author Response (AR1)

1. At the end of Intro, insert a paragraph outlining what this paper is trying to achieve and how the paper is structured

    (a) *The requested paragraph was added in Lines 133-138.*

2. Line 95: By Doppler velocity, do they mean the mean radial component?

    (a) *Corrected in Line 152.*

3. Line 115: What does 'visible signatures' mean? Can you quantify?

    (a) *We rephrased this statement in Lines 203.*

4. Line 128: The authors say "Based on the profiles of vertical velocity [V ], we propose a new variable: [gradV ].." – What about spectral width? Is this available from routine scans?

   (a) *The Spectral width variable was not available in the analysed radar datasets. We added an statement about this in Line 192.*

5. Figure 2: For the VP plots on the left side, the y-axis should go from 0 to 8 km to be consistent with the QVP plots. What about panel (j)? Why is the 0 to 1 km omitted?

   (a) *As described on Lines 188-190, data collected at vertical incidence is contaminated by spurious echoes. Still, we modified the plot so the y-axis is consistent on both sides, enabling a straightforward comparison.*

6. Line 144: Define 'normalised' at this point.

   (a) *We added further information in Lines 222–223.*

7. Line 147: should 'estimate' be 'detect'?

   (a) *Agreed and corrected.*

8. Line 148: What does 'enhancements that the ML bring-up into the variables' mean?

   (a) *We modified the text as outlined in Line 229–230.*

9. Line 154: By 'elevation' do they mean 'altitude a.g.l'?

   (a) *Corrected in Line 188.*

10. Lines 156–159: Grammar needs to be improved, and also the text is ambiguous; the sentence doesn't make much sense.

   (a) *The text was modified to improve its readability, as shown in Lines 242–246.*

11. Lines 163: convective events are associated with different microphysical processes so ML doesn't apply.

   (a) *Agreed and corrected in Line 254.*

12. Line 168: Doesn't the radar perform 'bird-bath' scans routinely?

   (a) *Yes, the bird-bath scans are the VPs. Please note that in Lines 277–278 we explained the need to know first the height of the melting level (ML) to apply a bias-correction to $Z_{DR}$.*

13. Line 168: The sentence beginning 'Hence the Zdr ..' requires much more clarification.

    (a) *We added further explanation about this variable in Lines 273–276.*

14. Section 3 is verbose, not very technical and not well-written at all. Please rewrite. Also explain clearly why the peaks in $Z_H$, $Z_{DR}$ and $\rho_{HV}$ are at different heights above ground level and explain the difference between BB and ML.

    (a) *We rewrote Section 3 to improve its readability. Also, we added further explanation and discussion regarding the peak heights in Lines 235-241. As mentioned, the difference between the profiles generated from our datasets and previous studies relies on the type of profiles and the average process for constructing the profiles. This can be seen in Revision Figures 1 and 2. The difference between melting level, melting layer and the bright band is now explained in Lines 19-25.*

15. Line 237: Once again, explicitly say how the normalisation is performed.

    (a) *The normalisation process is described in Lines 347–352.*

16. Explain how equations (2) and (3) were derived. If published elsewhere, then insert reference for the derivations.

    (a) *Clarification about the derivation process of Equations (2) and (3) is provided in Lines 388.*

17. Line 267: Explain/justify why the second derivative was chosen.

    (a) *Please note that this is discussed in lines 434-441, 490-491 and illustrated in Figure 5b.*

18. Line 293: "QVPs and VPs of Zh, as these variables measure similar properties of the raindrops" What does this mean?

    (a) *We added further explanation about this variable in Lines 503-505.*

19. Line 303: What does "resides on relative low values of reflectivity" mean?

    (a) *We rewrote this statement as outlined in Lines 516-518.*

20. What is the purpose of Section 5.1 if only the Z comparisons are given? It's not clear how it is relevant to the rest of the paper.

[Figure]

Revision Figure 1: VPs comparison regarding peak heights.

[Figure]

Revision Figure 2: QVPs comparison regarding peak heights.

(a) *This section is intended as a validation of the reflectivity QVPs. We believe it is necessary to assess the consistency between VPs and QVPs as the ML algorithm is based on the geometry of the profiles. But apart from $Z_H$, it is not possible to compare the figures of the other polarimetric variables due to the azimuthal averaging on the construction of the QVPs.*

21. Regarding Fig. 9: What does 'FL estimated' represent exactly, that is in relation to the radar BB (peaks in all the variables), and the 0 deg C isotherm level?

    (a) *At this point, we compared heights of the $0\,°C$ Wet-Bulb isotherms and the output of the algorithm (i.e. the top boundary in the enhanced profile) described in step 2.d (lines 424-427). We modified Figures 9 and 12 to clarify the scatter plot.*

22. What about attenuation corrections needed for Zh and Zdr? Were these applied?

    (a) *We are aware that rain attenuation is an error source for radar QPE in particular when using low-elevation scans and this is why the height of the melting level is essential to implement rain attenuation correction algorithms. For most of the PPI scans used in this analysis (90-deg scans and higher elevation scans at 9-deg elevation) rain attenuation at C-band was relatively small as demonstrated by the total differential phase shift. This is because the rain region for most of the profiles was below 3km in altitude a.g.l., which is equivalent to about 20km in range when using the 9-deg elevation scans. For this reason no attempt was made to correct for attenuation. This is now clarified in the paper in Lines 198-201.*

**2 Response to Anonymous Referee 2**

This study proposes an approach to estimate the freezing level (FL) using vertical/quasi–vertical profiles (VP/QVP) achieved from polarimetric radar observations. The proposed approach was applied to some selected events, and the estimated FLs were evaluated using radiosonde observations. Based on the evaluation results, the authors concluded that the combinations of ZH, $\rho_{HV}$, and the gradient of the velocity V, and ZH, $\rho_{HV}$, and ZDR for each VP and QVP method are the best predictors for the FL estimation. I think that the study was well–designed, and the focus and experimental details and results of the study are clearly addressed in the manuscript. However, I have a basic question about the utility of this study for radar QPE and additional comments/suggestions for some other aspects presented in this study.

*We thank the reviewer for the positive remarks and for the interesting feedback/discussion that helped to improve our work. Please note that considering the reviewers' comments, we modified the paper title to 'Detection of the melting level with polarimetric weather radar'. We modified the manuscript as outlined below, replying point by point in blue. The changes refer to the marked–up version of the manuscript.*

**Major comments:**

1. Utility of FL height. The authors discuss the necessity of FL information for radar–based applications (e.g., QPE) in Introduction. In my opinion, what is really useful for radar applications is to provide a range of the melting layer (ML), not just a single value of FL height itself (as this study mostly devoted to find the FL height) because mixed (liquid–solid) precipitation is usually located below the FL height, and this is a significant challenge e.g. for rainfall and attenuation estimation. I am wondering what specific applications require the estimated FL height. I think that a bottom height of the ML presented in Figures 10 and 13 is much more useful than the FL height itself because the majority of scattering and propagation theories can be applied only to the region below this height (liquid precipitation or pure rain region).

   (a) *We completely agree with the reviewer on the importance of accurate detection of the bottom of the melting layer as most of the QPE algorithms can only be applied in the rain region. Unfortunately, if the output of the algorithm is the bottom of the melting layer, it would be challenging to validate it using the radiosonde datasets or some other instrument. Hence, the proposed algorithm detects both the ML and the bottom of the melting layer based on the geometry of the profiles and the ML is validated using radiosonde data. Then, the bottom of the melting layer can be determined using a fixed thickness or by using the output of the algorithm, as shown in Figures 10 and 13.*

2. QVP. It is not clear if either time-averaged or instantaneous QVP is used in the proposed FL detection algorithm. I think that instantaneous QVP is not appropriate for the proposed algorithm because it could be affected by local storm structures (although it is derived from higher elevation angles) particularly for the ones near the radar. If the authors used time-averaged QVPs, they need to clarify it and define the averaging time window. It might be helpful for readers to understand the QVP method if the authors provide a brief description on the background and procedures to retrieve QVP from radar observations, rather than just referring to Ryzhkov et al. (2016).

   (a) *We are aware of the advantages of using time-averaged QVPs, and we did some tests using time-averaged QVPs. The algorithm considers this situation with the parameter k, which is helpful to deal with the smoothness caused by the time-averaging of the profiles, e.g. in Revision Figure 4, the profiles are averaged using a time-window of 30 minutes, and the parameter k is modified to allow the different values of the resulting profiles. The melting layer detected do not vary that much (see Revision Figures 3 and 4). Hence, we decided to display examples in the instantaneous QVPs format at this is the most common format of QVPs. We added further discussion about this Lines 602-609.*

3. FL spatial variability. I think that the proposed QVP method results in the average FL over the entire radar domain while the VP method yields limited FL to the radar site (if VP was obtained from a 90 degree elevation angle). I am wondering how the spatial variability of FL over the radar domain looks like, and the authors may compare the FL information retrieved from the NWP model with the one achieved from this study. It might be helpful to discuss this spatial variability issue in the discussion section as a limitation of this approach.

   (a) *A strong motivation for this work was to avoid relying on NWP products. One of the advantages of the algorithm is that it enables the estimation of the ML based entirely on the radar data; this is really helpful to implement corrections that depend on hydrometeor discrimination. We agree with the reviewer that there is a spatial variability of the ML over the radar domain. Still, after weighing the options, we considered that for the ML accuracy required in radar corrections, a straightforward algorithm and its validation using radiosonde surpass the complexity of data retrieved from numerical models and its computationally expensive runs, as showed by Hall et al. (2015) or Mittermaier and Illingworth (2003). We discussed the ML spatial variability in Lines 609-615.*

4. Error analysis. Whereas the analyses presented in this study focused on finding the best predictors of the polarimetric radar observations, it is

[Figure]

Revision Figure 3: Instantaneous QVPs and detected melting layer, related to a stratiform-type rain event.

[Figure]

Revision Figure 4: Time-averaged QVPs and detected melting layer, related to a stratiform-type rain event.

valuable to characterize the structure of errors resulted from the proposed methods. I think that it would be useful to demonstrate error distributions of each VP and QVP method (e.g., P14 and P26), rather than just reporting "the errors in the FL estimation using either VPs or QVPs are within 250m."

    (a) *We provide an error analysis in Figure 9 and 12 along with discussion in Section 6.*

**Minor comments:**

1. Line 10 Maybe better to remove "extremely."

    (a) *Corrected.*

2. Line 24–26 It would be interesting to compare the FL heights computed from between this study and the NWP model.

    (a) *Please refer to the answer of major comment No. 3.*

3. Line 87 Please define "UKMO."

    (a) *Corrected, UKMO refers to the UK Met Office.*

4. Line 106 Please replace "twice daily" with "twice a day."

    (a) *Corrected.*

5. Table 1 I think that the "Location" in Table 1 represents coordinates on a certain projected coordinate system. Geographic coordinates are more common and please provide latitude and longitude of the radar site.

    (a) *Agreed and corrected. Figure 1 and Table 1 are now in geographic coordinates.*

6. Figure 2 Please use consistent height (y-axis) and color scales for the same radar observables to enable easy comparisons between left and right panels for (a)–(h). Please also define "HTI" in the figure caption.

    (a) *Figure 2 was updated with consistent y-axis and similar color scales as requested.*

7. Line 144 Please clarify if QVPs shown in Figure 3 were time-averaged before they were normalized.

    (a) *Please check the answer provided to major comment No. 2.*

8. Line 181–182 How are "type of precipitation" and "phase of the hydrometeors" different?

(a) *We corrected this statement in Lines 279-282.*

9. Line 278 It turned out that "magnitude (k) of Ppeak" was a threshold (e.g., parameter) for peak magnitude (Line 287). Please clarify it here.

   (a) *We correct this aspect of the algorithm and clarify it throughout the manuscript.*

10. Line 291–294 Something is missing. Please rewrite.

    (a) *We rewrote this subsection to improve its readability.*

11. Line 300 Why do the authors compare VPs and QVPs? Is this comparison performed because the authors used instantaneous QVPs for FL estimation? I think that they (VP and QVP) are not necessarily consistent, and QVP should be used with timeaveraging to avoid local storm effects and capture the consistent vertical structure with VP.

    (a) *This section is somewhat intended as a validation of the construction of the QVPs. We consider it necessary to assess the consistency between both types of representations as the ML algorithm is based on the geometry of the profiles. But apart from $Z_H$, it is not possible to compare the figures of the polarimetric variables due to the azimuthal averaging on the construction of the QVPs. Please refer to Lines 602-609 for a detailed discussion on this matter.*

12. Section 5.2 This section does not describe the result of this study and should be moved to the "Methodology" section.

    (a) *Agreed and corrected.*

13. Line 352 Please replace "better" with "best."

    (a) *Corrected.*

14. Line 358 Why P16? Both ZH and [grad V] are the elements of P26. [grad V] was used for P16–P31, not just for P16.

    (a) *The purpose of showing the performance of $P_{16}$ in Figures 10c and 10d was to emphasise the value of the proposed variable $grad V$. We removed this analysis in the revised version of the manuscript.*

15. Line 359 Figures 10a and 10b instead of "Figures 13a and 13b?"

    (a) *We appreciate this observation, references are now correct.*

16. Line 361–362 The estimation procedure of the ML bottom was not described.

(a) *We added further description regarding the ML bottom detection in Lines 424-427 and in Figure 5b.*

17. Line 383 Please replace "better" with "best."

   (a) *Corrected.*

18. Line 386 Why P10? P10 does not have to be mentioned here because the two factors shown in Figure 13 are also included in P14.

   (a) *We used the variable $P_{10}$ to compare the different outputs of the algorithm. However, we removed this analysis in the revised version of the manuscript.*

19. Line 404–407 I think that the ZDR calibration bias is not an issue in this study because relative ZDR values (e.g., normalized) are used to construct vertical profiles. ZH also contains the calibration issue.

   (a) *We agree with the reviewer, the calibration in both, $Z_H$ and $Z_{DR}$ are not an issue when implementing the proposed ML identification algorithm. However, if we want to use $Z_{DR}$ quantitatively, then we must ensure $Z_{DR}$ is calibrated. Hence the necessity of the knowledge of the ML before the implementation of the $Z_{DR}$ calibration procedure. On the other hand, please note that $Z_H$ is routinely calibrated by the UK Met Office.*

**3 Response to Anonymous Referee 3**

The manuscript describes a technique to estimate the Freezing Level (FL) height, motivated by its practical use in downstream hydrological applications. The methods blend QVPs/VPs ideas as a convenient way to summarize the dual–polarization, vertical properties to inform this retrieval.

Overall, the manuscript accomplishes the application it sets out to perform. However, the effort seems limited in that it amends previous ideas with potentially questionable inputs. Such applications may be publishable within the scope of AMT, but this seems to require substantial edits (not a trivial re–write). The manuscript is long, yet not particularly organized in how it presents concepts, physical discussions. Most statements probably should be more conservative. It is not always clear what is original, or why this advancement matters? One radar advantage (somewhat lost with QVPs) is ability to capture FL variability spatially when compared to model output, surface, or radiosonde information. The manuscript claims originality from QVPs, but avoids when it is appropriate to use QVPs, e.g., important trade–offs for this decision. QVPs could be a smart substitution, but this choice encourages compensating errors. It is also not clear the outcome (i.e., FL estimates to match 0C Temperature) is the best target (i.e., 0C Wet Bulb Temperature)..

*We thank the referee for the detailed review. The comments were considered for the revised version of the paper. Please note that considering the reviewers' comments, we modified the paper title to 'Detection of the melting level with polarimetric weather radar'. In the following, we provide below point–by–point answers (in blue) to the comments.*

**Major comments:**

1. What accuracy does one require "FL" estimates, and is this important? What is the 'value–added', aka, why this specific approach? What is the advantage over existing ML ideas?

   (a) *The accuracy of ML estimates depends on the application. For instance, Kitchen et al. (1994) quoted 200m as the required accuracy in the ML height for VPR correction, whereas for rain attenuation correction the accuracy in the ML height could be lower (see Islam et al. (2014)). The added value of our algorithm is its capability to detect the ML height and at a certain degree, the melting layer, using data collected by operational weather radars. Previous studies require the processing of data collected/generated by other instruments, or data not available in operational radar networks. We believe that the proposed ML algorithm is helpful for radar data corrections that require the knowledge of the ML or the boundaries*

*of the ML, and when running NWP models it is not a feasible option.*

2. QVPs are spatial averages that favor widespread precipitation, homogeneous fields. QVPs are clever, convenient, but one 'practical' issue is related to their generation –e.g., this requires more statements on the tolerance for 'when' (under what conditions) these are generated, e.g., 'how frequently' does this result in useful retrievals? What about 'edge case' QVPs that may be generated, but require filtering? Basically, how confident are we that all conditions that allow a QVP are also equally viable as inputs?

   (a) *We believe that a thorough review of the construction process and limitations of the QVPs is out of the scope of this work and probably the subject of a new paper. Nevertheless, during the design of our ML algorithm, and looking at a large number of QVPs, we proposed different thresholds and parameters in the algorithm that are useful to identify the ML signatures from QVPs. In fact, we analysed a larger number of QVPs (almost one year of data) to identify potential problems during the implementation of the algorithm. This helped us to develop a robust algorithm. The results presented in this paper only cover events where both radiosonde and radar data were available. In the revised version of the paper, we added discussion on this matter in Lines 602-622*

3. QVP averaging removes ability to define regions of mixed precipitation (azimuthally) as one example issue common to FL/ML literature. The variability of the FL can be substantial, studies suggesting O[500 meters] – variability as large as the melting layer – and radar sectors where 'rain' switches to 'snow'. This argues QVPs are not suitably fine-grained, would struggle in locations where this is a concern, e.g., Boodoo et al. (2010).

   (a) *One of the main advantages of the QVP methodology is its ability to document the characteristics of the ML, as demonstrated by Griffin et al. (2018), Kaltenboeck and Ryzhkov (2017) or Ryzhkov et al. (2016). We are aware of the limitations of this methodology when large spatial variability of the ML is present in the PPI scans and this may only be mitigated using other inputs like data produced by numerical models, but previous studies e.g. (Hall et al., 2015) demonstrated the ability of radar measurements over numerical models to accurately detect the ML. Our algorithm only works with QVPs that are likely to contain ML signatures if the conditions outlined in the paper are satisfied. Please refer to Lines 602-622 for a detailed discussion of the ML spatial variability. Our goal is not to produce estimates of ML heights at every azimuth angle given the variability of the precipitation and noise in the radar measurements.*

*The QVPs smooth out to some extent this variability and the multiplication of the normalised profiles ensures that the ML estimation is robust as demonstrated when validating the ML estimates with radiosonde data.*

4. Melting onset is not at 0C temperature, rather at the 0CWet Bulb temperature. When viewing from radar, the height when one claims a melting response is typically lower, e.g., delay in measurement sensitivity to melting, but also the RH is often not 100%. A concern is if one becomes too interested in a retrieved 'match' to a radiosonde target that is not always correct. While the 0C temperature is the historical 'freezing level' definition, it is not the one (or only) hydrological applications care about associated with 'contaminated' radar signatures (e.g., 'bright band' shape also starts above with aggregation processes). This is partially why I suspect VP/velocity profiles are not as seeming useful in the offered, e.g., this is more a case of poor target/definitions than velocity not being a highly useful input.

(a) *We relied on the idea that the radar rain measurements are related to events with relative humidity near 100%, that is why the algorithm was designed to match the Dry-bulb $0\,°C$ measured by the radiosonde. Nevertheless, we agree with the reviewer on the necessity to analyse the relation between Dry-Bulb/Wet-Bulb temperature. In Revision Figure 5) we present an analysis through a year of radiosonde measurements and we found that, although the height is somewhat lower, the variation is not significant as theoretically, the Dry-Bulb and the Wet-Bulb temperatures are similar in the rain medium (as measured by the radar). Furthermore, we updated the error analysis shown in Figures 8-13 using $0\,°C$ Wet-Bulb isotherms heights.*

5. QVP-based dual-pol measurement profiles have different issues for interpretation; For example, ZDR profiles do not rapidly increase until onset of melting (0C Wet Bulb), whereas Z is increasing above the ML owing to aggregation. Unfortunately, where these signatures occur in altitude is complicated further when aggregation, melting are not the same spatially, then averaged in a QVP. The QVP issues are exacerbated when coupled with nonuniform beam-filling NBF issues that smear profiles. This calls into question concepts for 'combining' Z and RHOHV (or variants therein) – as in Wolfensberger et al. reference – as confusing when based on QVPs. It is not clear the order of operations, and how/when one averages, combines such fields. It makes a difference in the eventual input profile validity. Moreover, it may lead to solutions that 'work', but come to a matching answer for the wrong reasons.

(a) *We agree with the reviewer that certain microphysical process fingerprints may be reduced in magnitude because of the averaging*

[Figure]

Revision Figure 5: Heights of 0 Wet–bulb/Dry–bulb temperatures.

*process on the construction of the QVPs, as showed by Kumjian et al. (2016). However, we consider that the ML fingerprints are strong enough to surpass this limitation; this is why we analysed all the possible combination of polarimetric variables and compare them with the radiosonde data. The resultant profiles may not represent all the microphysical processes inside the profile, but a profile with enhanced ML signatures instead. Also, note that an altitude constrain was defined in the algorithm to minimise the effects of these problems, i.e. the algorithm is constrained to a height of 5 kilometres, and the QVPs are constructed from relative high elevation angles, in this case, for 9-deg scans, the height of the centre of the beam is similar to 30 km in range, which is relatively close to the radar and minimise the NBF problem.*

6. Effort is spent on explaining dual–polarization signatures (QVP), but the important process aspects are not too well described. Radar response to processes/properties (signatures) to include melting, aggregation, break–up/fallout, etc., are undoubtedly difficult. These processes and observed properties are smoothed/complicated further in response to known radar (system) bias, NBF, etc. Averaging and other processing details distort things further, esp. regions that preferentially feature different density or mass flux into these melting layers, RH, vertical motions, etc. There are a few resources for discussion on QVP signatures of the bright band (and reasons for its variability), e.g., Kumjian et al. (2016).
I call attention to nonuniform beam filling NBF in particular, and melting onset expectations. Illustrations for potential offsets in radar quantities

are found in Ryzhkov (2007). For intermediate tilts being used for QVPs, the expectation should be for modest biases in quantities owing to NBF (e.g., smearing) – This is complicated by the QVP averaging if the fields are not homogeneous. It is possible to model how well certain combinations of quantities may demonstrate compensating issues if one attempted to multiply those profiles, at different tilts, etc. – much of that would also arguably change on when those QVP averaging was performed (multiply before QVP, or QVP before multiply?). Again, it does not make sense (to this reviewer) how Z and RHOHV fields can be multiplied to generate a useful profile without factoring in several details (thus, also not surprised it may not apply consistently well, either).

(a) *We are aware of the microphysical processes related to rain events. Still, a detailed explanation of all of them based only on data collected from operational weather radar is beyond the scope of this paper. This task requires data collected from research radars with higher resolution that improves the rain process's understanding in a microphysical level. Instead, we describe the observed signatures related to the ML only, based on a long-term analysis of QVPs and VPs, as they are the algorithm's foundation. On the other hand, the NBF effect was analysed, and a threshold in the range is proposed to mitigate their effects. Moreover, the normalisation process and the algorithm's design help to mitigate the effects of the beam broadening, as the algorithm does not detect the ML using quantitative values of polarimetric variables, but the strong gradients that the ML generate in the profiles. This is why we consider that the rationale behind the algorithm's design is justified: by multiplying normalised profiles, we want to generate a new profile with enhanced gradients related to the ML, and to some extent, washout other peaks that difficult the implementation of a peak-finder algorithm.*

**Minor comments:**

1. The 'freezing level' is a poor term, persists in operations. Perhaps 'melting level', as frozen media begins to melt at that level.

   (a) *We agree with the reviewer, "Detection of the melting level with polarimetric weather radar" could be a more appropriate title for the paper. We added further discussion on differences between these terms in the manuscript.*

2. The authors use examples for the QVP, VP profiles in several figures. Critically, I find these examples often physically nonintuitive, even when the authors imply these as only meant as examples. For example, one expects the Z peak to be higher in altitude (above) of the ZDR peak, with the ZDR and RHOHV peaks located at similar altitudes. If the authors

retain the physical discussions on the dual–polarization signatures, the reasons for such relative behaviors are perhaps more important. These are also far less commonly described.

(a) *Please note that the behaviours described by the reviewer are mainly related to profiles extracted from individual slant ranges. Similar profiles can also be observed in our datasets, e.g. in Revision Figures 1 and 2 the peak in $Z_H$ is higher in altitude than the peaks of $Z_{DR}$ and $\rho_{HV}$ for the azimuthal profiles (orange lines). However, due to the averaging process carried out its construction, the peaks' height of VPs and QVPs (blue lines) show behaviours that differ from previous studies based on RHI scans or theoretical profiles. We provide an explanation of these profiles in Sections 3 and 6.*

3. I had an impression velocity gradient ideas were being presented as novel/unique. The authors should likely consult the profiling radar literature (e.g., works of C. Williams, other profiling radar echo classification manuscripts) that commonly use gradients of mean Doppler velocity in their efforts. As above, I suspect velocity is more accurate / informative profile input (when available) for assessing the wet bulb zero for reasons of its improved vertical resolution and sensitivity to its relative 'change point' with melting onset. I suspect open–code / python change point / inflection techniques would also apply vertically as compared to gradient ideas, too.

(a) *We are aware of the work related to the profiling radar literature, e.g. (Tian et al., 2019; Williams et al., 1995, 2005, 2007). Still, we consider that the velocity gradient was not used as an input variable to delimit the melting layer. We'll appreciate it if the author could provide more references on this matter. On the other hand, even though that we are aware of the use of inflexion techniques and the use of the Velocity as an input of some algorithms (we even made some experiments using "raw" velocity profiles) we conclude and demonstrate that the use of the second derivative (or in this case, its complement $1 - gradV$) fits better into the design of our algorithm.*

**References**

Griffin, E. M., Schuur, T. J., and Ryzhkov, A. V. (2018). A polarimetric analysis of ice microphysical processes in snow, using quasi-vertical profiles. *Journal of Applied Meteorology and Climatology*, 57(1):31–50.

Hall, W., Rico-Ramirez, M. A., and Krämer, S. (2015). Classification and correction of the bright band using an operational C-band polarimetric radar. *Journal of Hydrology*, 531:248–258.

Islam, T., Rico-Ramirez, M. A., Han, D., and Srivastava, P. K. (2014). Sensitivity associated with bright band/melting layer location on radar reflectivity correction for attenuation at C-band using differential propagation phase measurements. *Atmospheric Research*, 135–136:143–158.

Kaltenboeck, R. and Ryzhkov, A. (2017). A freezing rain storm explored with a C-band polarimetric weather radar using the QVP methodology. *Meteorologische Zeitschrift*, 26(2):207–222.

Kitchen, M., Brown, R., and Davies, A. G. (1994). Real-time correction of weather radar data for the effects of bright band, range and orographic growth in widespread precipitation. *Quarterly Journal of the Royal Meteorological Society*, 120(519):1231–1254.

Kumjian, M. R., Mishra, S., Giangrande, S. E., Toto, T., Ryzhkov, A. V., and Bansemer, A. (2016). Polarimetric radar and aircraft observations of saggy bright bands during mc3e. *Journal of Geophysical Research: Atmospheres*, 121(7):3584–3607.

Mittermaier, M. P. and Illingworth, A. J. (2003). Comparison of model-derived and radar-observed freezing-level heights: Implications for vertical reflectivity profile-correction schemes. *Quarterly Journal of the Royal Meteorological Society*, 129(587 PART A):83–95.

Ryzhkov, A., Zhang, P., Reeves, H., Kumjian, M., Tschallener, T., Trömel, S., and Simmer, C. (2016). Quasi-Vertical Profiles—A New Way to Look at Polarimetric Radar Data. *Journal of Atmospheric and Oceanic Technology*, 33(3):551–562.

Tian, J., Dong, X., Xi, B., Williams, C. R., and Wu, P. (2019). Estimation of liquid water path below the melting layer in stratiform precipitation systems using radar measurements during MC3E. *Atmospheric Measurement Techniques*, 12(7):3743–3759.

Williams, C. R., Ecklund, W. L., and Gage, K. S. (01 Oct. 1995). Classification of precipitating clouds in the tropics using 915-mhz wind profilers. *Journal of Atmospheric and Oceanic Technology*, 12(5):996 – 1012.

Williams, C. R., Gage, K. S., Clark, W., and Kucera, P. (01 Jul. 2005). Monitoring the reflectivity calibration of a scanning radar using a profiling radar and a disdrometer. *Journal of Atmospheric and Oceanic Technology*, 22(7):1004 – 1018.

Williams, C. R., White, A. B., Gage, K. S., and Ralph, F. M. (01 May. 2007). Vertical structure of precipitation and related microphysics observed by noaa profilers and trmm during name 2004. *Journal of Climate*, 20(9):1693 – 1712.

---

## Author Response (AR2)

**Final Response to Referees and Associate Editor on "Detection of the melting level with polarimetric weather radar"**

Daniel Sanchez-Rivas and Miguel Angel Rico-Ramirez

March 2021

**1 Report: Anonymous Referee #1**

*We thank the reviewer for the positive remarks and the interesting feedback/discussion that helped improve our work.*

**2 Report: Anonymous Referee #3**

*We are glad to see that we have met the reviewer's acceptance for publication. Following the reviewer's comments, we modified Sections 6 and 7 to improve its clarity. We also carried on a careful review of the manuscript to remove typos and other minor issues. Finally, we would like to thank the reviewer for the detailed comments on the review of the paper that helped to improve its quality and readability.*

**3 Response to Associate Editor.**

*We thank the editor for the positive remarks and guidance throughout the review process. We modified the manuscript as outlined below.*

**List of relevant changes in manuscript:**

1. We updated Figure 1 to meet the required copyright and distribution licence statements. Also, minor corrections to Figure1 labels were applied.

2. P.1, line 64 - p.2, line 12. We consider that moving the paragraph to this new location improves the section's clarity.

3. We addressed the reviewers' suggestions to break down large paragraphs in Section 6 and modified the format in Section 7, adding a bullet list of the conclusions.

4. Several typos and other minor issues were fixed.